# Oxa-Iboga alkaloids lack cardiac risk and disrupt opioid use in animal models

Václav Havel [1,16], Andrew C. Kruegel[1,16], Benjamin Bechand[1,16], Scot McIntosh[2], Leia Stallings[2], Alana Hodges[2], Madalee G. Wulf[1], Mel Nelson[3,4,5], Amanda Hunkele[6,7], Michael Ansonoff[8], John E. Pintar [8,9], Christopher Hwu[1], Rohini S. Ople[7,10], Najah Abi-Gerges [11], Saheem A. Zaidi[12,13], Vsevolod Katritch [12,13], Mu Yang [14], Jonathan A. Javitch [3,4,5], Susruta Majumdar [6,7,10], Scott E. Hemby[2] & Dalibor Sames [1,15] ✉

Ibogaine and its main metabolite noribogaine provide important molecular prototypes for markedly different treatment of substance use disorders and co-morbid mental health illnesses. However, these compounds present a cardiac safety risk and a highly complex molecular mechanism. We introduce a class of iboga alkaloids – termed oxa-iboga – defined as benzofuran-containing iboga analogs and created via structural editing of the iboga skeleton. The oxa-iboga compounds lack the proarrhythmic adverse effects of ibogaine and noribogaine in primary human cardiomyocytes and show superior efficacy in animal models of opioid use disorder in male rats. They act as potent kappa opioid receptor agonists in vitro and in vivo, but exhibit atypical behavioral features compared to standard kappa opioid agonists. Oxa-noribogaine induces long-lasting suppression of morphine, heroin, and fentanyl intake after a single dose or a short treatment regimen, reversal of persistent opioid-induced hyperalgesia, and suppression of opioid drug seeking in rodent relapse models. As such, oxa-iboga compounds represent mechanistically distinct iboga analogs with therapeutic potential.

Ibogaine is the major psychoactive alkaloid found in the iboga plant (*Tabernanthe iboga*), a shrub native to West Central Africa[1]. While the root has been harvested as a ceremonial and healing commodity in Africa for centuries, the use of the iboga plant or pure ibogaine has recently become a world-wide movement, with growing numbers of ibogaine healers, providers, and clinics, largely driven by the crises of drug addiction, trauma, despair, and spiritual starvation[2–4]. Ibogaine induces psychedelic effects that typically include dream-like states (oneiric effects, oneirogen), panoramic and interactive memory recall, experiences of death and rebirth, confrontation with personal trauma, and loosening of maladaptive habits[5]. Ibogaine is unique among other psychedelics for its ability to rapidly interrupt opioid drug dependence, as measured by a dramatic reduction in opioid withdrawal symptoms[6]. Although rigorous demonstration of clinical efficacy via

controlled clinical trials is pending, the profound anti-addiction effects of ibogaine have been amply documented in anecdotal reports and open-label clinical trials, including rapid and long-lasting relief of drug cravings, increased duration of abstinence, as well as long-term reduction of anxious and depressive symptoms in subjects with drug dependence, PTSD, and traumatic brain injury (TBI)[7–10]. The clinical claims of ibogaine's therapeutic properties have been recapitulated in numerous rodent models of substance use disorders (SUDs) and depression[11–14].

Ibogaine has a complex chemical structure, in which the tryptamine motif is intricately embedded in the isoquinuclidine ring, leading to a polycyclic tryptamine system that defines the iboga alkaloids (Fig. 1a, d)[15]. The pharmacology of ibogaine is also complex, induced by ibogaine and its main metabolite noribogaine - the dominant

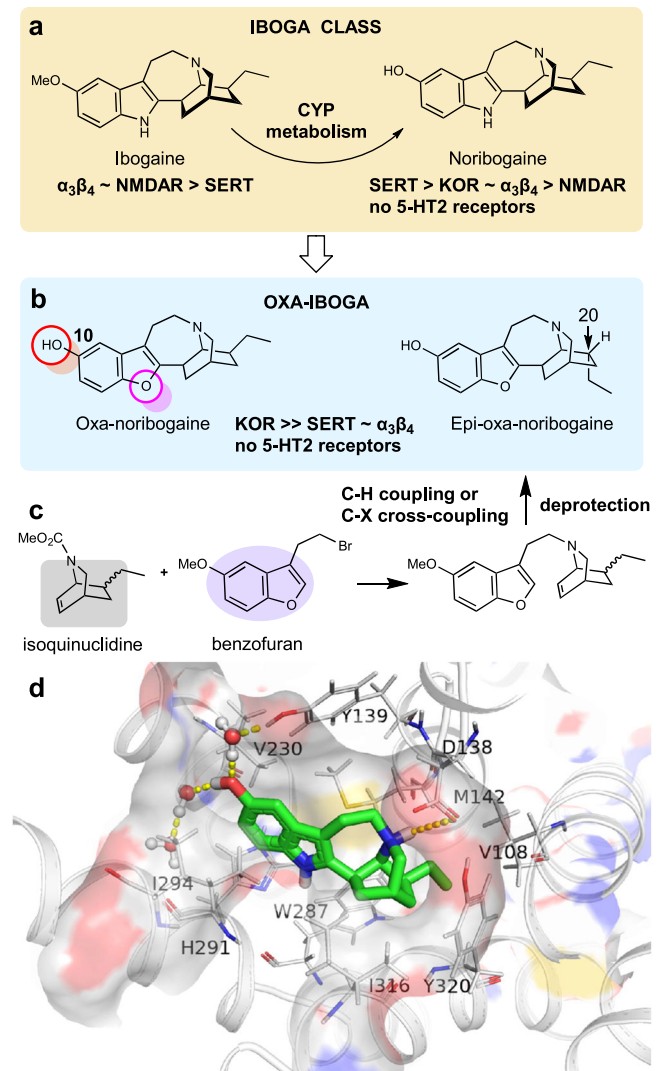

**Fig. 1 | Oxa-iboga is a distinct class of iboga alkaloids discovered by structural editing of ibogaine, enabled by efficient de novo chemical synthesis. a** Ibogaine is a substance with broad therapeutic effects and complex pharmacology, that is distinct from classical psychedelic tryptamines. Relative potencies at known molecular targets are shown. **b** Oxa-iboga analogs are defined by the replacement of indole with benzofuran, resulting in accentuation of the KOR activity on the iboga pharmacological background. **c** De novo synthesis of iboga molecular framework rests on the catalytic union of the two main structural components of oxa-iboga skeleton, the isoquinuclidine and benzofuran ring systems. **d** Docking pose of noribogaine (carbon frame in green) in sticks representation inside KOR structure (active receptor state). Hydrogen bonding near the C10 phenol and tertiary amine are highlighted by yellow dashed lines. KOR (kappa opioid receptor), α3β4 (α3β4 nicotinic acetylcholine receptors), SERT (serotonin transporter), 5-HT (serotonin), NMDAR (N-methyl-D-aspartate receptor).

circulating species. The known molecular targets for both ibogaine and noribogaine indicate a complex profile of multimodal polypharmacology, characterized by modulation of many molecular targets and signaling pathways via several different modes of action. These include, for example, N-methyl-D-aspartate receptor (NMDAR, ion channel blocker), α3β4 nicotinic receptor (α3β4 nAChR, antagonist), serotonin transporter (SERT, inhibitor and pharmacochaperone), and kappa opioid receptor (KOR, agonist)[16–19]. Mechanistically, ibogaine thus appears to bridge several different classes of psychoactive substances, including the anesthetic psychedelics (NMDAR blockers such as phencyclidine (PCP) or ketamine), kappa opioid receptor

agonists (such as salvinorin A or U50,488), and monoamine reuptake inhibitors (such as imipramine), but shows no direct interaction with the 5-hydroxytryptamine type 2 receptors (5-HT2R), setting ibogaine apart from the classical psychedelics (Fig. 1a, b, Supplementary Fig. S3). We have developed synthetic methods for de novo synthesis of the iboga alkaloid scaffold, which has unlocked a broad exploration of its structure and pharmacology (Fig. 1c)[18,20]. Specifically, we developed and optimized both nickel-catalyzed and palladium-catalyzed processes for 7-membered ring formation, which enable modular preparation of the benzofuran-containing iboga analogs (Supplementary Fig. S1, S2, including X-ray crystallographic structure). These analogs show greatly potentiated KOR activity on top of the iboga pharmacological background, constituting a distinct class of iboga alkaloids (Fig. 1, Supplementary Fig. S3).

## Results

### Oxa-iboga analogs are KOR agonists with partial receptor signaling efficacy

Noribogaine, the starting and comparison point for our studies, acts as a KOR partial agonist in a bioluminescence resonance energy transfer (BRET) assay for G protein activation (rat KOR, $EC_{50} = 6.1\,\mu M$, $E_{max} = 52\%$, Fig. 2b, Supplementary Fig. S5), consistent with previous reports[21]. We found that substitution of the indole NH group in noribogaine with oxygen dramatically accentuates KOR activity: oxa-noribogaine (oxa-noriboga, Fig. 1b) binds to the KOR (mKOR, $K_i = 36\,nM$) and activates G protein in the $[^{35}S]GTP\gamma S$ assay (mKOR, $EC_{50} = 49\,nM$, $E_{max} = 92\%$) and the BRET assay (rKOR, $EC_{50} = 43\,nM$, $E_{max} = 82\%$), representing a large increase in potency versus noribogaine (Fig. 2b, Supplementary Fig. S5). Oxa-noribogaine is more than ten-fold selective for KOR versus MOR and about seven-fold over DOR in the $[^{35}S]GTP\gamma S$ binding assay using mouse opioid receptors (Supplementary Fig. S4). No binding affinity for nociceptin receptor (ORL-1/NOP) was detected for noribogaine or oxa-iboga compounds up to 10 μM (Supplementary Fig. S3e and Table S1). To determine whether the KOR signaling initiated by oxa-noribogaine is comparable to a full KOR agonist, we examined seven relevant Gα protein isoforms in a BRET assay for activation of G protein heterotrimers (TRUPATH), which indicates attenuated signaling efficacy across four Gα isoforms compared to the standard KOR agonist, U50,488 (Fig. 2c)[22,23].

We also used a recruitment assay based on the Nb33 nanobody that senses an active conformation of the KOR, thus limiting the signal amplification of G protein activation assays and approximating true intrinsic signaling efficacy of tested compounds (BRET assay utilizing a luciferase-tagged KOR and Venus-tagged Nb33, Fig. 2d)[24,25]. It has previously been shown using the Nb33 sensor that U50,488 has comparable signaling efficacy to the endogenous KOR ligand, dynorphin A[26]. In direct comparison to U50,488 as the standard, oxa-iboga compounds exhibit markedly reduced signaling efficacy in this assay ($E_{max(oxa-noriboga)} = 52\%$, $E_{max(epi-oxa)} = 34\%$), differentiating themselves from classical KOR agonists such as U50,488. We finally examined the oxa-iboga compounds in a β-arrestin2 recruitment assay (Fig. 2e), showing once again markedly reduced signaling efficacy of these compounds compared to U50,488. These studies show that oxa-noribogaine induces partial, intermediate signaling efficacy; namely reduced efficacy compared to full KOR agonists such as U50,488, but greater efficacy than the low efficacy agonist noribogaine. We also examined the signaling of oxa-iboga compounds at mouse MOR using two functional readouts analogous to the KOR; the BRET G protein activation ($E_{max(oxa-noriboga)} = 81\%$, $E_{max(epi-oxa)} = 28\%$) and BRET Nb33 sensor assays ($E_{max(oxa-noriboga)} = 44\%$, $E_{max(epi-oxa)} =$ inactive/below detection limit, Fig. S5), showing markedly lower signaling efficacy induced by epi-oxa versus oxa-noribogaine. This finding provides a potential explanatory model for the different contribution of MOR to the antinociceptive activity in mice of the two compounds, where epi-oxa's effect is entirely driven by KOR while that of oxa-

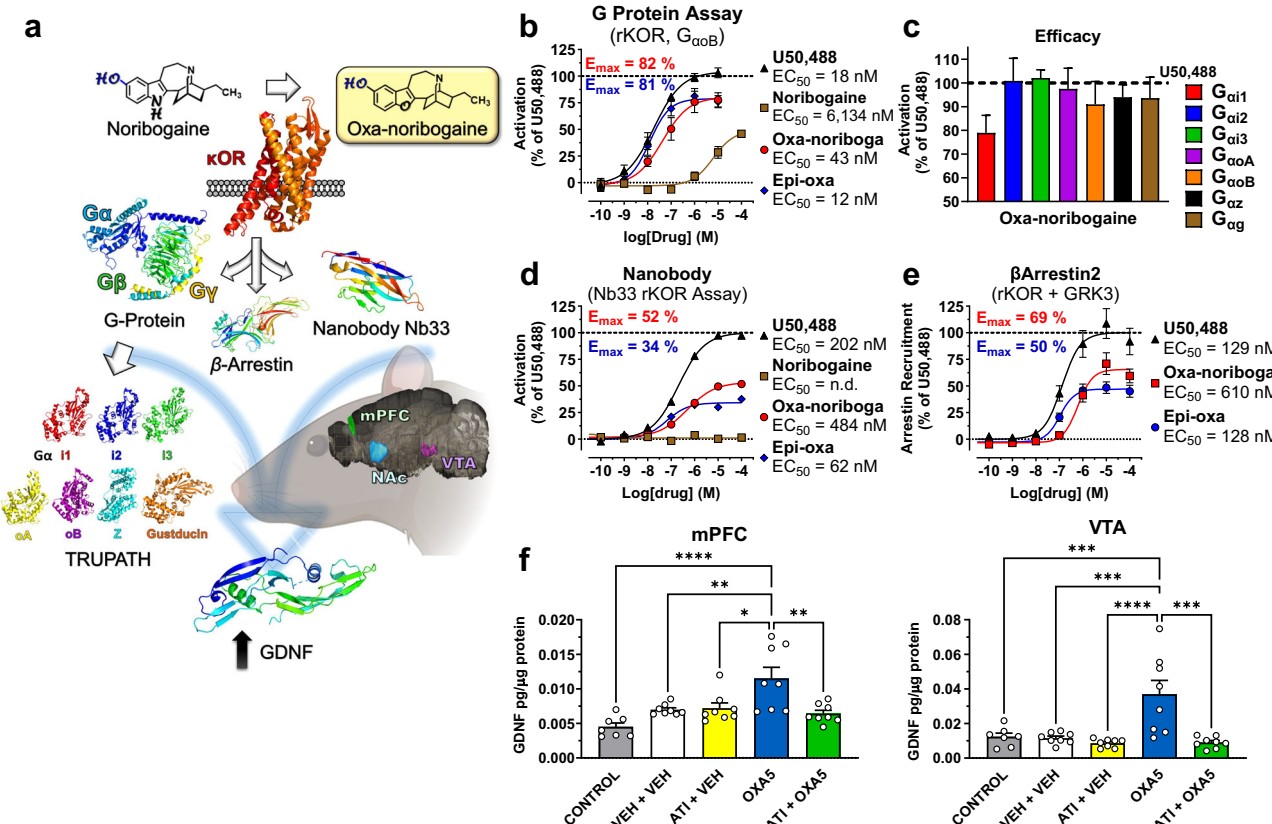

**Fig. 2 | KOR dependent upstream and downstream molecular signaling pathway effects of oxa-iboga compounds. a** Illustration of the KOR-induced molecular signaling pathway assays in vitro and in vivo. The location and shape of medial prefrontal cortex (mPFC), nucleus accumbens (NAc) and ventral tegmental area (VTA) are highlighted in a rodent brain - representative illustration of a left hemisphere (sagittal slice, mouse, Allen Institute[99]). **b** Oxa-iboga compounds are agonists of rat KOR in vitro, as demonstrated by a G protein activation BRET assay. **c** Oxa-noribogaine displays signaling efficacy across Gα isoforms, efficacy compared to U50,488 in the TRUPATH BRET assay. Pooled data ($n = 3$ biological replicates) for each subunit were analyzed using a nonlinear regression and the calculated span ± SEM parameters are presented (dose response curves Supplementary Fig. S6). **d** In the nanobody Nb33 sensor recruitment assay, which approximates true intrinsic signaling efficacy, oxa-noribogaine analogs are partial agonists. **e** Additionally, markedly reduced signaling efficacy for oxa-noriboga

analogs was detected in the β-arrestin2 recruitment assay (KOR results on panels **b**, **d**, **e** are presented as mean ± SEM, $n = 3$ biological replicates). **f** Administration of oxa-noribogaine (40 mg/kg; i.p.) significantly increased GDNF protein levels in the mPFC and VTA after 5 days (OXA5). Pre-treatment of rats with KOR selective antagonist aticaprant (ATI, 1 mg/kg s.c.) before oxa-noribogaine administration prevents the increase of GDNF expression in mPFC and VTA. mPFC: CONTROL - OXA5 ($P < 0.0001$), VEH + VEH - OXA5 ($P = 0.0078$), VEH + ATI - OXA5 ($P = 0.0123$), OXA5 - ATI + OXA5 ($P = 0.0022$) and VTA: CONTROL - OXA5 ($P = 0.0010$), VEH + VEH - OXA5 ($P = 0.0004$), VEH + ATI - OXA5 ($P < 0.0001$), OXA5 - ATI + OXA5 ($P = 0.0001$). Experiments were conducted using separate groups of animals (Control $n = 7$, all other groups $n = 8$ subjects). Data are presented as mean ± SEM. Statistical test used: One-way ANOVA with Tukey's multiple comparisons test, *$P < 0.05$, **$P < 0.01$, ***$P < 0.001$, ****$P < 0.0001$. Source data are provided in the Source Data file.

noribogaine has small MOR contribution (see in vivo pharmacology below and Discussion).

The phenol group in the 10-position is the essential component of both the iboga and oxa-iboga KOR pharmacophores, as agonist activity is lost if the phenol is masked or removed (Supplementary Table S1). Noribogaine's binding pose obtained by docking to the active KOR state identified water-mediated hydrogen-bonding interactions between the C10 phenolic hydroxyl and tyrosine 139[3.33], as well as an ionic interaction between the isoquinuclidine amine and acidic aspartate 138[3.32] (Fig. 1d). The KOR potency and selectivity can be further modulated by the inversion of the geometry of the C20 center, such as in the endo-epimer, epi-oxa-noribogaine (epi-oxa, Fig. 1b), although the effects are relatively modest (Fig. 2b, d, e). However, the docking studies do not provide an explanation for the large differences in binding and signaling potencies between noribogaine and oxa-noribogaine as the docking poses are similar (Fig. S4e–g).

For the rest of iboga pharmacology, SERT is arguably the most rigorously validated molecular target of noribogaine[19,27]. The oxa-iboga compounds largely maintain 5-HT reuptake inhibitory activity with a modest loss of potency compared to noribogaine

($IC_{50(oxa-noriboga)} = 711$ nM versus $IC_{50(noribogaine)} = 286$ nM, Supplementary Fig. S3a). Similarly, the inhibitory activity of oxa-noribogaine at the α3β4 nicotinic receptors and NMDA receptors is comparable to that of noribogaine, with nearly a 2-fold increase in potency (α3β4 hAChR, $IC_{50(oxa-noribogaine)} = 2.9$ μM vs $IC_{50(noribogaine)} = 5.0$ μM; NMDAR, $IC_{50(oxa-noribogaine)} = 24$ μM vs $IC_{50(noribogaine)} = 42$ μM Supplementary Fig. S3b, c). A broad receptor screen showed a favorable pharmacological profile (Supplementary Fig. S3e) with more than 100-fold separation in binding potency between KOR and non-opioid human targets (with the exception of SERT and α3β4 molecular targets, Supplementary Fig. S3a–c).

## Oxa-noribogaine increases protein levels of neurotrophic factors in specific brain regions

We next examined down-stream molecular signaling targets, focusing on neurotrophic factors as mediators of neuroplasticity and potentially long-lasting therapeutic effects. We centered on glial cell line derived neurotrophic factor (GDNF, Fig. 2a, f) as it has been implicated in ibogaine's and noribogaine's suppression of alcohol intake in drinking rats[28], and shown to be induced dose-dependently by

ibogaine in wild type unconditioned rats[29]. Although this neurotrophic factor plays complex roles in the neurobiology of substance use disorders[30,31], targeted elevation of GDNF protein levels in specific brain areas, such as the ventral tegmental area (VTA), contributes to the attenuation or reversal of various aspects of addiction-related effects (e.g. drug intake, reward expression, and dopamine neuron firing patterns). Several brain regions were selected for analysis, including the VTA, whose dopaminergic neurons project to the medial prefrontal cortex (mPFC), where deep layer pyramidal cells provide reciprocal projections to the VTA, which in turn provides dense dopaminergic innervation in the nucleus accumbens (NAc, Fig. 2a); all three brain areas and the corresponding circuits play prominent roles in addiction biology. We focused directly on alternations of neurotrophin protein levels in these brain regions as 1) GDNF gene expression loci do not always match those of GDNF receptors, and 2) GDNF can be transported over long distances. Specifically, we investigated long-term changes in levels of GDNF protein in the VTA, NAc, and mPFC up to five days after a single dose of oxa-noribogaine (Fig. 2f). Remarkably, we observed a 200% increase in GDNF levels in the VTA and 100% in the mPFC on day 5 after treatment. To test the association of GDNF levels in the VTA and mPFC with KOR activation, we treated rats with aticaprant, a selective KOR antagonist, prior to oxa-noribogaine administration. Under these conditions, the induction of GDNF was blocked in both mPFC and VTA, revealing protein levels comparable to those of control/vehicle rats (Fig. 2f). These results indicate that the long-term induction of GDNF by oxa-noribogaine in relevant brain circuits is a down-stream consequence of KOR activation.

## Oxa-noribogaine induces potent antinociception, no rewarding or aversive behavior, and no pro-depressive-like effects

Oxa-noribogaine acts as a partial KOR agonist in terms of receptor signaling pathways (see above), and therefore, we examined the effects of KOR activation by this class of compounds in vivo. Oxa-noribogaine induced potent antinociceptive effects in male mice in the tail-flick assay (thermal nociception), with comparable potency to the standard KOR agonist, U50,488 (Fig. 3a, $ED_{50(oxa-noriboga)} = 3.0$ mg/kg vs $ED_{50(U50)} = 2.2$ mg/kg; s.c. administration). In KOR knock-out (KOR-KO) male mice, the analgesic effect was substantially attenuated, with a one-log-unit right shift in the dose response curve ($ED_{50(WT)} = 2.3$ mg/kg vs $ED_{50(KOR-KO)} = 17.7$ mg/kg, Fig. 3b). Using MOR knock-out (MOR-KO) male mice, a small rightward shift in the dose-response curve was observed (Fig. 3b, $ED_{50(MOR-KO)} = 4.3$ mg/kg). Epi-oxa-noribogaine also induced a potent antinociceptive effect, which was abolished in KOR-KO mice, while a marginal rightward shift was observed in MOR-KO mice (Fig. 3c, $ED_{50(WT)} = 1.3$ mg/kg, $ED_{50(KOR-KO)} =$ n.a., $ED_{50(MOR-KO)} = 2.0$ mg/kg). In tail-flick in wild type female mice, epi-oxa exhibited a rightward shift in potency compared to males ($ED_{50(male)} = 1.9$ mg/kg vs $ED_{50female)} = 9.7$ mg/kg, Fig. 3a, d), as did U50,488 ($ED_{50(male)} = 2.2$ mg/kg vs $ED_{50female)} = 6.3$ mg/kg, Fig. 3a, d), consistent with previous reports for typical KOR agonists, including U50,488[32,33]. In contrast, oxa-noribogaine showed a small rightward shift in the tail-flick assay in one cohort ($ED_{50(male)} = 3.0$ mg/kg to $ED_{50(female)} = 4.9$ mg/kg, Fig. 3a, d), with practically no difference in other cohorts used as controls for KOR-KO experiments ($ED_{50(male)} = 2.3$ mg/kg vs $ED_{50(female)} = 2.0$ mg/kg, Fig. 3b, e). In female knock-out mice, the analgesic effect of oxa-noribogaine was notably diminished in the KOR-KO and to a lower degree in MOR-KO genetic models ($ED_{50(WT)} = 2.0$ mg/kg vs $ED_{50(KOR-KO)} = 12.4$ mg/kg vs $ED_{50(MOR-KO)} = 7.1$ mg/kg, Fig. 3e). Thus, the analgesic potency and efficacy of oxa-noribogaine is comparable between male and female mice and primarily driven by KOR activation in both sexes. Pharmacological studies with opioid receptor antagonists supported this interpretation and showed that delta opioid receptors (DOR) do not contribute to the analgesic effects of this class

(Supplementary Fig. S7a). The antinociceptive test not only indicates a desirable therapeutic-like effect, but also provides a useful physiological readout for functional KOR engagement in vivo (pharmacodynamic readout), which enables dosing calibration for behavioral studies.

It is well established that KOR agonists induce dose-dependent hallucinosis in humans, accompanied by sedation, aversive effects and mood worsening in healthy subjects[34]. In rodents and other species, kappa agonists also induce a sedation-like phenotype characterized by reduced locomotor activity without a complete loss of ambulatory functions (animals respond to gentle physical stimulation, but appear sedated when left undisturbed). At nearly maximal analgesic doses (e.g., $ED_{80}$, 5.4 mg/kg), oxa-noribogaine did not induce sedation in either male or female mice, indicated by locomotor activity comparable to that of the vehicle group in the open field (OF) test (Fig. 3f–h). In stark contrast, epi-oxa induced a strong sedative effect at an equianalgesic dose, an effect completely reversed by the KOR antagonist aticaprant (Fig. 3h). Thus, these two diastereomers, while producing analgesia with similar potency, show differences in sedative-like effects in mice. However, at supra-analgesic doses of oxa-noribogaine ( > 10 mg/kg, s.c.), sedation in mice is observed. Thus oxa-noribogaine provides a differential dosing window between analgesic and sedative effects, in contrast to epi-oxa and typical kappa agonists such as U50,488, where the dose-dependence of sedation approximately follows that of analgesia (Fig. 3g).

We next assessed the rewarding or aversive effects of oxa-noribogaine using the place conditioning assay. Typical KOR agonists mediate stress effects and show aversive effects, or conditioned place aversion (CPA), while the majority of drugs of abuse produce conditioned place preferences (CPP) in this test[35]. Oxa-noribogaine showed no CPA or CPP at supra-analgesic doses in male mice, indicating no aversion or reward compared to significant conditioned place preferences for both cocaine ($P = 0.0029$) and morphine ($P = 0.001$, Fig. 3j). As sex differences have been reported for KOR modulators in reward-related behaviors in rodents[33], we also tested female mice and showed no rewarding or aversive effects induced by oxa-noribogaine (Fig. 3j). Thus, oxa-noribogaine does not function as an aversive stimulus in mice of either sex. Typical kappa agonists also induce worsening of mood in healthy humans and pro-depressive-like effects in rodents in the forced swim test[34,36]. Oxa-noribogaine demonstrated no acute depressive-like effects in male mice at highly analgesic doses (Fig. 3i). These results indicate that even at high functional engagement of KOR, pro-depressive or aversive behavioral effects are not induced by this compound.

A pharmacokinetic profile of systemically administered oxa-noribogaine was investigated in mice (Fig. 3k) and rats (Supplementary Fig. S7h). Oxa-noribogaine was brain penetrant and the estimated free drug concentrations in the brain after analgesic doses match well with the in vivo physiological readouts and in vitro pharmacological parameters (Supplementary Fig. S7e and Methods Section).

In summary, the in vitro and in vivo pharmacology indicate that oxa-noribogaine is an atypical KOR agonist that exerts a partial agonistic activity in upstream receptor signaling pathways (intermediate efficacy agonist) but potent and maximal analgesia (full agonist in vivo) in the absence of common side effects of kappa agonists, namely, acute aversion and pro-depressive behaviors.

## Oxa-iboga compounds do not show pro-arrhythmia risks in adult primary human heart cells

The use of ibogaine has been associated with severe cardiac side effects, most notably, cardiac arrhythmias and sudden death in humans. It has been suggested that these adverse effects are linked to the inhibition of human ether-a-go-go-related gene (hERG) potassium channels by both ibogaine and noribogaine[37]. hERG inhibition can result in retardation of cardiomyocyte action potential repolarization

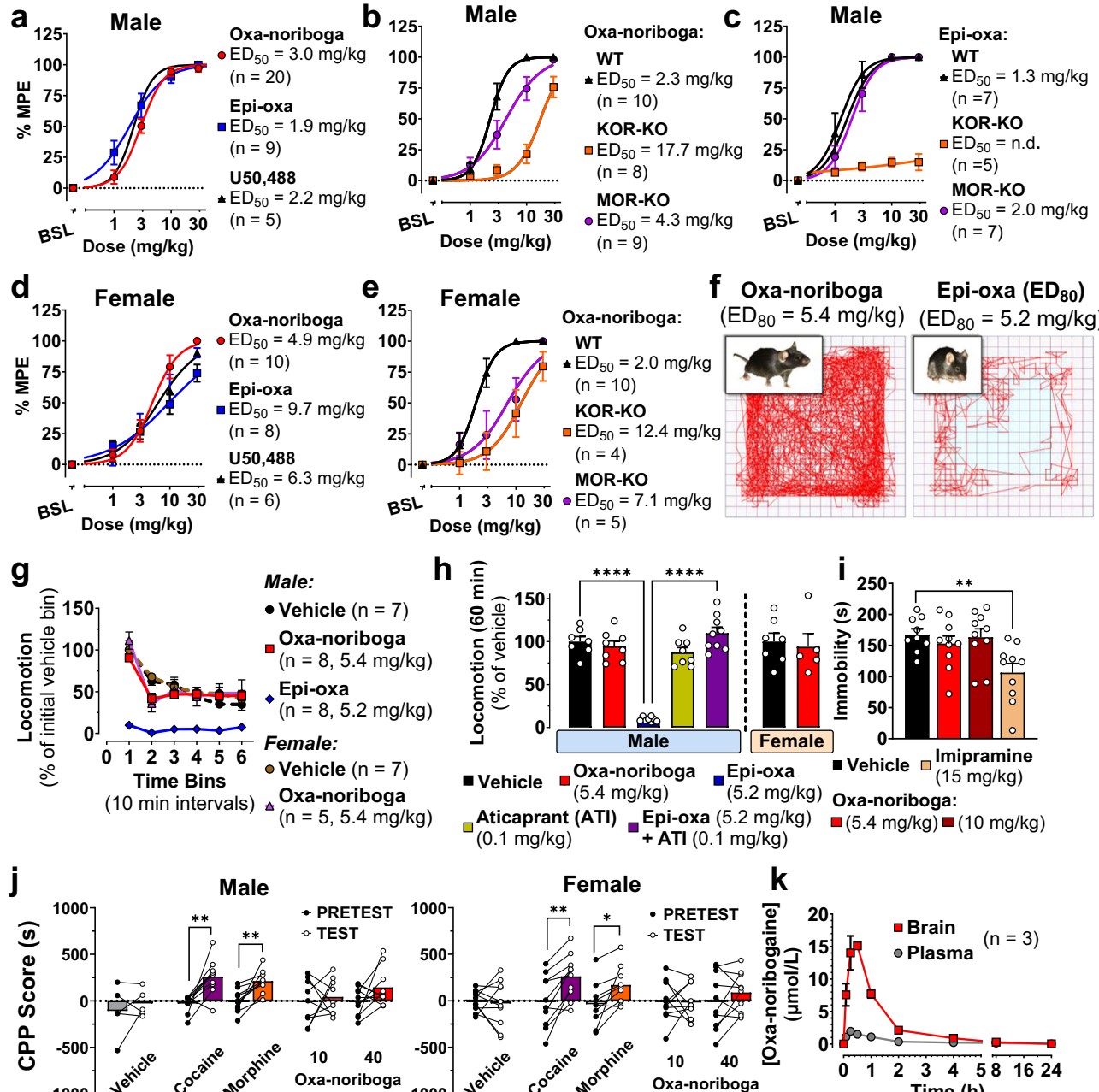

**Fig. 3 | Oxa-noribogaine is an atypical KOR agonist in vivo with no aversion and no pro-depression-like effects at efficacious analgesic doses. a** Oxa-iboga analogs induce potent analgesia in the mouse tail-flick test (male mice), comparable in potency and efficacy to the standard kappa agonist, U50,488. **b** Analgesia of oxa-noribogaine and epi-oxa-noribogaine (**c**) is KOR dependent as demonstrated in KOR knock-out mice (KOR-KO), compared to mu receptor knock-out (MOR-KO) and wild type (WT) mice (KOR-KO vs WT, $P < 0.0001$). **d** Oxa-noribogaine induces potent analgesia in female mice, primarily driven by KOR as demonstrated in **e** female KOR-KO mice compared to MOR-KO and WT. **f** Traces visualizing ambulatory distance traveled by WT mice in open field test (OF) show different effects by the isomers at equianalgesic doses. **g** Quantification of OF test (ED80 doses) for oxa-noribogaine (5.4 mg/kg, both male and female) and epi-oxa-noribogaine (5.2 mg/kg), data are normalized to initial locomotion of vehicle group. **h** Sedation of epi-oxa-noribogaine (5.2 mg/kg, $P < 0.0001$) is KOR-driven as demonstrated by pre-treatment ($P < 0.0001$) by the selective KOR antagonist aticaprant (ATI, 0.1 mg/

kg). Total locomotion over 60 min period is normalized to vehicle. **i** No pro-depressive-like effects were detected using the forced swim test after oxa-noribogaine administration (30 min post administration). Imipramine was used as a positive effect control ($P = 0.0031$). **j** Male and female mice do not develop a conditioned place preference or aversion (CPP/CPA) after administration of oxa-noribogaine ($P = 0.4531$, $P = 0.5994$, respectively), in contrast to cocaine (10 mg/kg: $P = 0.0029$ and $0.0013$) and morphine (20 mg/kg: $P = 0.0010$ and $0.0153$).
**k** Pharmacokinetic distribution of oxa-noribogaine in mice plasma and brain tissue (10 mg/kg). All drugs were administered via subcutaneous (s.c.) route except for intraperitoneal (i.p.) route for CPP/CPA test. % MPE (percentage of maximum potential effect). Data are presented as mean ± SEM. Statistical tests used: Unpaired (**h**, **i**) and paired (**j**) $t$-test, two-tailed, One-way (**i**) and Two-way (**g**, **j**) ANOVA with Šidák multiple comparisons test, *$P < 0.05$, **$P < 0.01$, ***$P < 0.001$, ****$P < 0.0001$. Source data are provided in the Source Data file.

and prolongation of the QT interval in the electrocardiogram, thus increasing the risk of arrhythmias[38]. This hypothesis is supported by a recent dose escalation clinical study, which reported a linear relationship between QT interval prolongation and plasma concentration of noribogaine (although this study may have been confounded by residual methadone)[39].

Preclinical assessment of the cardiotoxicity of experimental compounds is complicated by species differences in cardiac ion channel expression and pharmacology. Further, inhibition of the hERG channel alone is not sufficient to predict delayed ventricular repolarization and cardiac pro-arrhythmia risk, as modulation of other ion channels involved in different phases of the cardiac action potential may mitigate or exacerbate the QT prolongation/pro-arrhythmia risk (for example by inhibition of L-type calcium channels), and there may be species differences in ion channel pharmacology[40]. This is particularly relevant for compounds like iboga alkaloids with complex pharmacology. As a result, preclinical in vivo tests in rodents or other non-human species may be misleading.

We therefore used adult human primary cardiomyocytes in a state-of-the-art assay with high predictive validity for clinical cardiac effects[40]. Adult human primary ventricular myocytes are isolated from ethically consented donor hearts, field stimulated to induce stable contractility transients, and test compounds are both applied and their effects observed (Fig. 4). The assay detects pro-arrhythmic events such as after-contractions and contraction failures and has been validated with clinically characterized drugs including pro-arrhythmic drugs and non-pro-arrhythmic drugs[40]. The assay has high sensitivity and specificity, and outperforms many other assays due to the phenotypic stability of adult primary cardiomyocytes compared to human stem cell-derived cardiomyocytes[40,41].

Based on available noribogaine clinical data, we hypothesized that plasma concentrations above 300–400 nM would be associated with a considerable pro-arrhythmia risk in this assay[39]. As such, we tested this drug in the clinically relevant concentration range (0.1–10 µM, for more information see Supplementary Fig. S7, S8 and the Methods section). As expected, noribogaine showed a concentration-dependent pro-arrhythmia risk in the human cardiomyocyte assay, eliciting an increased frequency of aftercontractions and contraction failures (Fig. 4b, c). Greater than 20% incidence of any of the arrhythmia-associated events indicates a considerable pro-arrhythmia risk, which was observed at noribogaine concentrations of 1 µM or higher. In contrast, oxa-noribogaine and epi-oxa-noribogaine showed no pro-arrhythmic potential at any of the concentrations tested (up to 10 µM), despite the fact that oxa-noribogaine binds to hERG in the similar potency range as noribogaine ($K_{i(oxa\text{-}noriboga)}$ = 2.1 µM, Supplementary Fig. S3d vs $K_{i\ (noriboga)}$ = 2.0 µM, lit. value[37]). We hypothesize that the observed differences in pro-arrhythmia risks between the oxa-iboga and iboga compounds are related to the activity at multiple cardiac ion channels of the former compounds, likely engaging channels known to compensate hERG-mediated pro-arrhythmia risks (such as the channel Nav1.5). However, to elucidate the relevant mechanism will require detailed follow-up studies, which are beyond the scope of this report.

## Oxa-noribogaine suppresses morphine, heroin, and fentanyl self-administration, cue-induced reinstatement of opioid responding, and opioid-induced hyperalgesia

To assess the therapeutic potential of oxa-noribogaine, we utilized the rat intravenous self-administration (SA) paradigm, a widely accepted model of opioid use[42]. In this behavioral assay, animals are trained to respond on an operandum under a defined schedule of reinforcement to receive an intravenous infusion of a drug while discrete stimuli (tone and light) are paired with the drug delivery (Fig. 5a). For the purpose of direct comparison to the iboga alkaloids, we chose noribogaine as the standard control on the basis of these points: 1) it is the dominant and long circulating molecular species after ibogaine's administration in both humans and rats[14,27], 2) noribogaine shows nearly identical effects to ibogaine in the rodent opioid self-administration paradigm[11], and 3) noribogaine is structurally a close analog of oxa-noribogaine, differing by a single atom substitution. The specific experimental design (Fig. 5b, Methods section) was selected to replicate previous results[11] and validate noribogaine as the comparison drug in our experiments. The selected dose of noribogaine (40 mg/kg, i.p.) has previously been shown to fully suppress morphine SA acutely[11], and corresponds to the lower end of ibogaine psychedelic reset doses or flood doses used in ibogaine clinics (estimate based on simple allometric scaling).

We found that noribogaine (40 mg/kg, i.p.) suppressed morphine self-administration in the session immediately following administration (day 1) and led to a partial, but statistically significant, suppression of morphine intake on days 2 and 3, returning to pre-treatment level of morphine intake on day 5 (Fig. 5b). Using a non-drug reinforcer, we found responding maintained by food was also suppressed on day 1 (by >80%) but returned to baseline responding on Day 2 (Fig. 5c). These results replicate the previous studies in terms of the efficacy (extent of morphine intake suppression), selectivity (drug vs food), and duration of the effect[11]. When tested under the same conditions, oxa-noribogaine (40 mg/kg, i.p.) induced a more profound and longer-lasting suppression of morphine intake (Fig. 5b, P = 0.0001), an effect that was statistically significant for 7 days and showed observable suppression trends for at least 15 days (Supplementary Fig. S9a). One subject from this cohort showed a >80% decrease in morphine intake at the end of the second week (Days 12–15, Supplementary Fig. S9b). Following oxa-noribogaine administration, food-maintained responding was suppressed on day 1, but largely returned on day 2, and did not differentiate statistically from vehicle from day 3 onward (Fig. 5c). Thus, the suppression effect of a reset dose of oxa-noribogaine is morphine-specific from day 3 onward. We next examined a 10 mg/kg dose of oxa-noribogaine, which induced a strong acute suppression of morphine intake (>85% reduction), an effect that is morphine selective as there was only a small, non-significant effect on acute food responding (Fig. 5b, c). Morphine self-administration largely returned to baseline levels on subsequent days. Thus, a single 10 mg/kg dose enabled selective acute suppression of morphine self-administration without strongly affecting behavior motivated by natural rewards.

Illicit fentanyl use has become a major issue exacerbating the opioid epidemic and driving the alarming numbers of drug overdose deaths[43]. Fentanyl's high in vivo potency, brain penetration, and intrinsic efficacy at MOR, as well as its rapid onset of effects, likely contribute to its abuse potential and overdose rates. Although rigorous clinical comparative data are not available for the relative reinforcing efficacy of fentanyl compared to other opioids, anecdotal reports suggest a treatment-resistant phenotype in fentanyl-dependent users[44]. We, therefore, examined the activity of oxa-noribogaine in fentanyl SA under the SA procedure used for morphine. Remarkably, a single reset dose of oxa-noribogaine reduced the intake of fentanyl for at least five days (Fig. 5d), indicating the efficacy of this compound against fentanyl.

We next examined in more detail the potential behavioral mechanisms underlying the opioid intake suppression by oxa-noribogaine. To this end, we evaluated the dose-dependent effects of oxa-noribogaine on a portion of the morphine dose-effect curve that encompassed the ascending and descending portions of the function (peak of infusions at 20 µg/infusion, Fig. 5e). Specifically, we examined two oxa-noribogaine doses (in approximate half-log steps, 3 and 10 mg/kg) across all morphine infusion doses, to find that oxa-noribogaine decreased the intake of morphine at all doses, essentially flattening the morphine curve (Day 1, Fig. 5e, left panel). In addition to the acute effects, we also examined the effect of each oxa-noribogaine dose for five days following administration. At the morphine dose maintaining peak intake, we observed a clear, dose-dependent

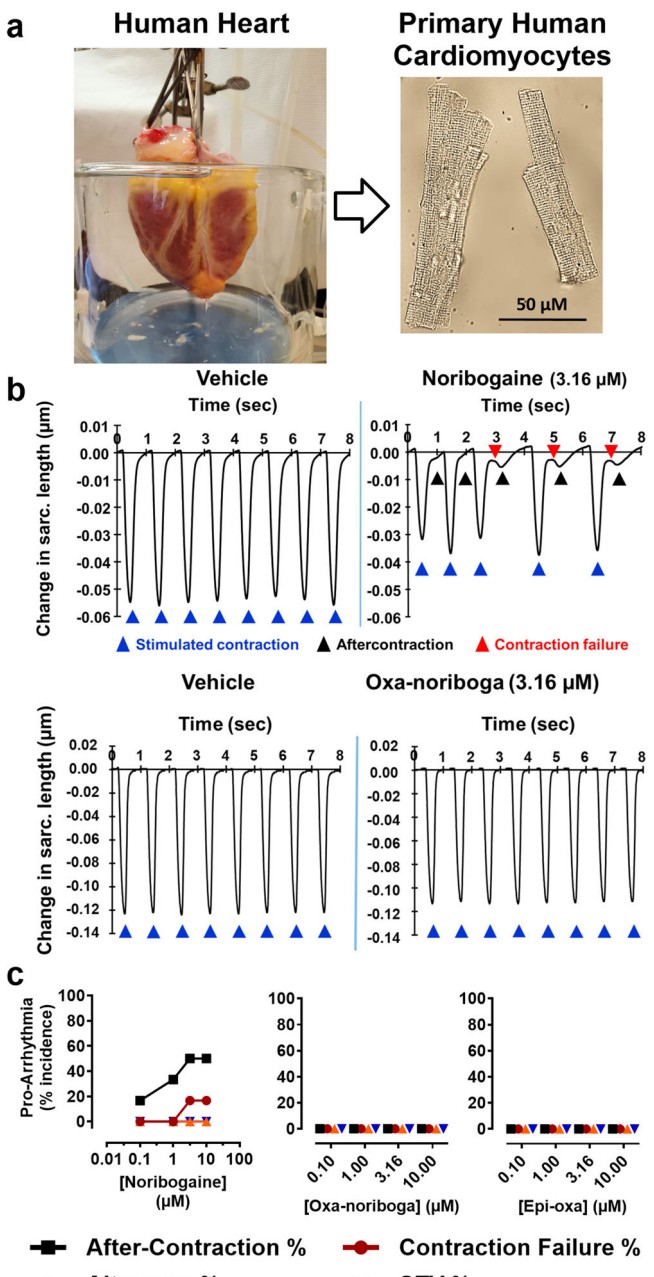

**Fig. 4 | Oxa-iboga analogs do not show pro-arrhythmia risk in adult human primary cardiomyocytes. a** Human heart is digested to isolate single cardiomyocytes, which are field stimulated to produce a regular pattern of contractility transients. These cells are phenotypically stable and provide a robust preclinical assay with high translational validity. **b** Representative traces capturing contraction-induced change in sarcomere length after administration of vehicle, noribogaine and oxa-noribogaine solutions. **c** Noribogaine demonstrates pro-arrhythmic potential by causing after-contractions and contraction failures in cardiomyocytes as quantified in the plot, providing validation of this assay for the iboga compounds. No pro-arrhythmic potential was detected for oxa- or epi-oxa-noribogaine (for donor information and replicates see Supplementary Fig. S8). Source data are provided in the Source Data file. Alternans % (percentage of repetitive alternating short and long contractility amplitude transients), STV % (Short-Term Variability in percentage).

attenuation of morphine SA lasting over at least five daily operant sessions after each oxa-noribogaine dose (Fig. 5f). Once again, we observed a lasting suppression of morphine intake that effectively flattened the morphine dose-response function, even days after oxa-

noribogaine administration. These results indicate that oxa-noribogaine reduces morphine responding both acutely (indirect-antagonist-like effect, drug-on effect) and long after oxa-noribogaine is cleared from the subjects (drug-off effect).

Next, we examined the dose-response of oxa-noribogaine on SA of heroin, another opioid drug with high abuse potential and a long history of use. These results showed a dose-dependent suppression of heroin infusions (4.5 μg/infusion) both acutely and post-acutely, with a measurable suppression even after 10 mg/kg dose (Fig. 5g), indicating reproducibility of these effects across cohorts and opioid drugs.

Relapse, the return to drug use following a period of abstinence, is one of the most significant challenges in the treatment of substance use disorders and therefore, a critical dimension to assess in the development of pharmacotherapies for OUDs. Cue-induced reinstatement of responding is a well-established assay of drug seeking, consisting of a period of self-administration followed by extinction of responding in the absence of drug and associated stimuli. For the test session, responding on the previously drug-associated lever is recorded in the presence of stimuli previously associated with drug delivery. For fentanyl, robust baseline responding was extinguished over 8 days, followed by cue-induced reinstatement in which responses were comparable to the pre-extinction level in the vehicle cohort (Fig. 6a). In contrast, oxa-noribogaine (10 mg/kg) reduced reinstatement responding by approximately 60%, whereas there was no effect on inactive levers, suggesting a drug cue-specific effect. Similarly, in another cohort, morphine reinstatement was also strongly reduced by oxa-noribogaine, at a dose that has minimal effect on food responding (10 mg/kg).

Finally, we examined the effect of oxa-noribogaine on opioid-induced hyperalgesia (OIH) in rats induced by chronic exposure to morphine and measured by mechanical hypersensitivity (von Frey test). Chronic opioid use leads to increased sensitivity to nociception and this pain state, known as hyperalgesia, is linked to physical dependence and is a part of the protracted withdrawal syndrome[45]. Post opioid cessation, OIH can last for several weeks in rodents and months in humans, likely contributing to a lasting negative affect and propensity to relapse[46]. Subcutaneous implantation of morphine pellets in rats for eight days produced a robust OIH phenotype that persisted for at least one week post implant removal (Fig. 6b, $P = 0.004$, 8 h post-implant removal, and $P = 0.0157$ one-week post-implant removal). One dose of oxa-noribogaine (30 mg/kg, i.p.) induced a complete reversal of the OIH phenotype 6 h post oxa-noribogaine administration and attenuation one week later. These findings indicate a lasting alleviation of persistent pain states in opioid dependent and post-dependent rats after one dose of oxa-noribogaine.

## Discussion

Chronic use of opioids and other reinforcing drugs can lead to persistent negative alterations of neurocircuitry that underlie addiction-related states in animals, and presumably, in humans[47,48]. Ibogaine serves as inspiration for pharmacotherapeutic agents capable of inducing profound and lasting interruption of drug addiction phenotypes, as demonstrated by reduced withdrawal symptoms, drug-intake, drug-seeking, and relapse in human and animal subjects[7]. The current molecular hypothesis for the rapid and lasting therapeutic effects invokes induction of neurotrophic factors, such as GDNF and BDNF, that drive desirable neuroplasticity and neuro-restorative alterations of relevant circuitry[28,29]. However, due to the vast pharmacological complexity of ibogaine and noribogaine, it remains unclear which molecular targets initiate these downstream molecular, circuit-centered, psychological, and behavioral changes.

We here describe the oxa-iboga compounds as a distinct class of iboga alkaloids created by a single strategic structural permutation of the iboga skeleton, which leads to several major consequences. In comparison to noribogaine these include: 1) attenuated cardiotoxicity

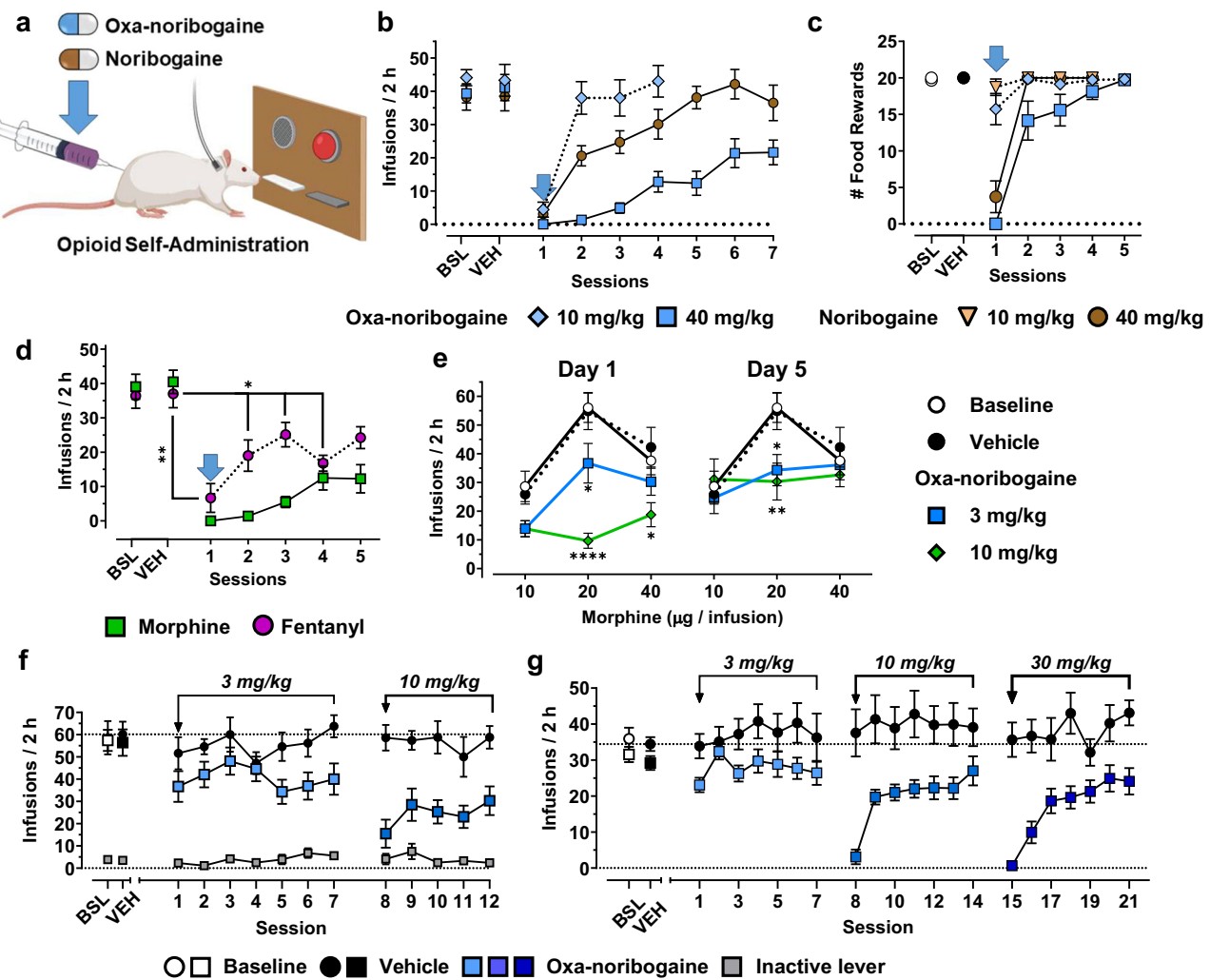

**Fig. 5 | Oxa-noribogaine induces acute and long-lasting suppression of morphine, heroin, and fentanyl self-administration in rats. a** A schematic depiction of the experimental treatment paradigm of opioid use (male rats used in this study). **b** Oxa-noribogaine (40 mg/kg) is more efficacious than noribogaine (40 mg/kg) in suppressing morphine self-administration (*P* = 0.0001, seven sessions). One injection of oxa-noribogaine (40 mg/kg) results in statistically significant suppression of morphine intake for 7 days. The effect on morphine self-administration was dose dependent. **c** Food operant intake (natural reward) is reduced following administration of 40 mg/kg but not 10 mg/kg (*P* < 0.0001). The moderate dose (10 mg/kg) has a marginal effect on food intake (all study groups *n* = 7). **d** One injection of oxa-noribogaine (40 mg/kg, *n* = 9) results in statistically significant suppression of fentanyl intake for 4 days (comparison with morphine data from Fig. 5b). **e** Visualization of acute (Day 1) and post-acute (Day 5) dose-dependent effects of oxa-noribogaine (3 and 10 mg/kg) on morphine dose-effect function in morphine self-administration (10 µg/inf, *n* = 8; 20 µg/inf, *n* = 10; 40 µg/inf, *n* = 8). On Day 1, oxa-noribogaine dose dependently decreases self-administration of morphine 20 µg/inf

(oxa-noribogaine doses, 3 mg/kg: *P* = 0.0419 and 10 mg/kg: *P* < 0.0001) and 40 µg/inf (10 mg/kg: *P* = 0.0327), and intake continued to be significantly decreased on Day 5 for 20 µg/inf (Oxa-noribogaine doses, 3 mg/kg: *P* = 0.0243 and 10 mg/kg: *P* = 0.0044). **f** A temporal profile across all sessions of the dose dependent effects of oxa-noribogaine in the rat cohort self-administering 20 µg/inf (*n* = 10) of morphine (active lever – shades of blue, inactive – grey). Control parallel cohort received injections of vehicle (*n* = 5) solution. **g** Repeated escalating dosing of oxa-noribogaine (3, 10 and 30 mg/kg; *n* = 11) enable acute suppression of heroin (4.5 µg/infusion) intake (10 mg/kg, session 8: *P* = 0.0012 and 30 mg/kg, session 15: *P* < 0.0001) and the long term suppression effect propagates up to 7 days post last intervention (session 16: *P* = 0.0009, session 18: *P* = 0.0137 and session 21: *P* = 0.0234). All drugs were administered via intraperitoneal (i.p.) route. Data are presented as mean ± SEM. Statistical tests used: Unpaired (**b**, **g**) and Paired (**c**, **e**) *t*-test, two-tailed, One-way (**c**) and Two-way (**b**, **c**, **d**, **e**, **g**), repeated measures ANOVA or Mixed Model with Šidák multiple comparisons test *\*P* < 0.05, **\*\*P* < 0.01, \*\*\**P* < 0.001, \*\*\*\**P* < 0.0001. Source data are provided in the Source Data file.

2) enhanced therapeutic-like activity in rodent behaviors with relevance to OUD, and 3) increased KOR agonistic activity, providing a mechanistically more tractable agent.

The oxa-iboga compounds act as potent partial KOR agonists in vitro, but exhibit atypical behavioral features compared to standard KOR agonists, such as a lack of aversion or pro-depressive effects. The present studies show a clear differentiation of oxa-iboga from standard KOR agonists via markedly decreased receptor signaling efficacy in several functional readouts that assay intrinsic signaling efficacy of the receptor, activation of seven Gα isoforms, and recruitment of β-arrestin2. Oxa-noribogaine does not show any clear preference

between Gi activation and β-arrestin2 recruitment, but instead shows lower signaling efficacy across these molecular pathways compared to a full agonist. We therefore propose the partial signaling efficacy as a working hypothesis for at least some of the unique behavioral effects of oxa-noribogaine; similar to the rationale for increased therapeutic index of partial MOR agonists[24,49], or the signaling efficacy hypothesis for the non-hallucinogenic 5-HT2A agonists[50,51]. Despite its lower signaling efficacy upstream, oxa-noribogaine induces robust KOR-dependent elevation of GDNF protein levels in the VTA and mPFC for at least five days after a single administration. Further, oxa-noribogaine is a fully efficacious analgesic in a thermal antinociception

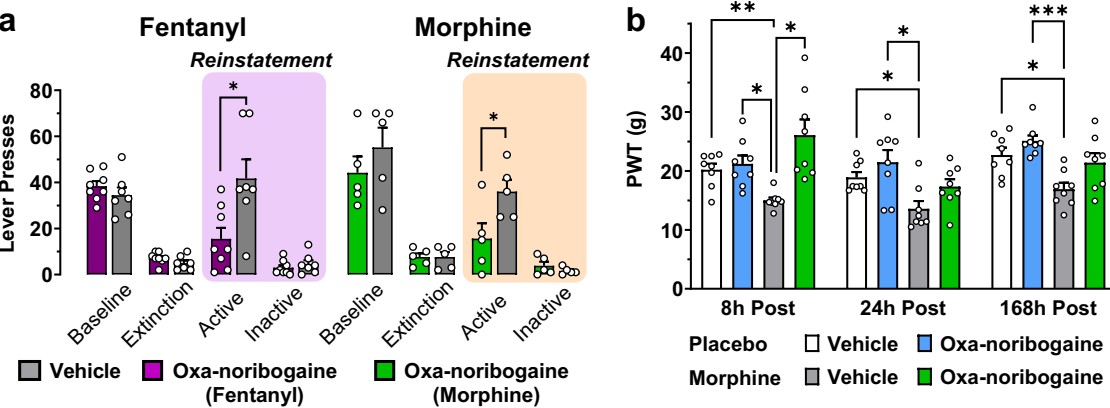

**Fig. 6 | Acute suppression of reinstatement to opioid responding and long-lasting alleviation of opioid-induced hyperalgesia in rats by a single dose of oxa-noribogaine. a** A single dose of oxa-noribogaine (10 mg/kg, i.p.) reduces cue-induced fentanyl ($P = 0.0141$, $n = 8$) and morphine ($P = 0.042$, $n = 5$) seeking. **b** Robust OIH response resulting from a chronic exposure to morphine, measured as paw withdrawal threshold (PWT; Von Frey assay), persisting for up to 7 days (Placebo-Veh vs. Morphine-Veh: 8 h $P = 0.004$, 24 h $P = 0.0258$ and 168 h $P = 0.0157$)

is suppressed by a single dose of oxa-noribogaine (30 mg/kg, $n = 8$), Placebo-Oxa vs. Morphine-Veh (8 h $P = 0.0127$, 24 h $P = 0.0305$ and 168 h $P = 0.0003$) and Morphine-Veh vs. Morphine-OXA (8 h $P = 0.0164$). Data are presented as mean ± SEM. Statistical tests used: Unpaired t-test, two-tailed (**a**) and Two-way repeated measures ANOVA with Tukey's multiple comparisons test (**b**), $*P < 0.05$, $**P < 0.01$, $***P < 0.001$, $****P < 0.0001$. Source data are provided in the Source Data file.

assay in mice, while alleviating OIH pain states in opioid-dependent and post-dependent rats. Oxa-noribogaine reduces intake of morphine, heroin, and fentanyl, as well as cue-induced reinstatement responding for fentanyl and morphine in rats.

However, we caution that the partial KOR agonism rationale is only correlative, as we have not directly examined the downstream signaling processes and their causal relationship to the specific behavioral readouts. It is therefore unclear which amplification mechanisms are in play that may enable a partial agonist at the KOR-G protein level to induce full-agonist-like antinociceptive effects while lacking KOR-driven aversion. Alternative hypotheses should be considered for the atypical effects of the oxa-iboga compounds (e.g., lack of aversion or pro-depressive effects), such that these compounds may preferentially target KOR-MOR or other KOR-GPCR heterodimers, and or KOR activation interacts with the complex iboga background pharmacology. These represent important mechanistic questions that are beyond the scope of this study and will be pursued in future investigations. However, the signaling efficacy differences at MOR, where epi-oxa shows markedly reduced in vitro efficacy compared to oxa-noribogaine, provide an explanatory model for the different contributions of MOR to the KOR-driven analgesic effects of the two oxa-iboga isomers: epi-oxa (no MOR contribution) and oxa-noribogaine (minor contribution). Thus, we propose that there is a signaling efficacy threshold at KOR/MOR that needs to be reached, to result in a measurable in vivo pharmacodynamic readout. The MOR signaling efficacy/activity is too low for epi-oxa to contribute to the compound's antinociceptive effects.

With regard to the preclinical OUD studies, however, there is the question of predictive validity of the animal models for candidate therapeutics[52,53]. In the present study, the starting point is a drug with profound clinical effects (although not yet confirmed in controlled studies), which was reverse translated to provide a benchmark for preclinical evaluation of new compounds, in terms of both efficacy and safety. In direct comparison to noribogaine, oxa-noribogaine is superior in terms of both maximal efficacy (extent of intake suppression) and duration of its after-effects. For another comparison, 18-methoxycoronaridine (18MC) is a synthetic iboga analog that has progressed through preclinical and Phase I clinical safety studies, but its clinical efficacy has not yet been reported. In preclinical assays in rats, 18MC showed efficacy and duration of effect on morphine intake comparable to ibogaine and noribogaine[54], and thus, an oxa-iboga

analog may offer a superior iboga prototype for SUD treatment. Ibogaine has also served, in part, as inspiration for the creation of 1,2,3,4,5,6-hexahydroazepino[4,5-b]indoles, for example compound U-22394A ("U" for Upjohn Company, also known as PNU-22394) that has been in clinical tests but not for SUD indications[55]. An analog of PNU-compounds (named TBG) has recently been examined in a heroin SA test in rats and showed no lasting effect on heroin SA beyond the acute and non-selective suppression of heroin and sucrose intake (a long lasting effect was observed in a reinstatement readout)[56]. However, the PNU-compounds are pharmacologically distinct from iboga, as they are direct ligands of multiple 5-HT receptors and lack activity at KOR and other known iboga targets, while iboga alkaloids have no direct interactions with 5-HT GPCR receptors within a relevant concentration range. The PNU-compounds are structurally and pharmacologically related to lorcaserin (5-HT2C > 5-HT2A preferring agonist), which prior to its market withdrawal, generated excitement in SUD research (e.g., showing suppression of heroin SA in rats[57] and cocaine SA in monkeys[58]). Lorcaserin maintained effective suppression of cocaine SA for an extended period of time by daily dosing prior to SA sessions (14 days); however, the suppression effect completely dissipated as soon as the treatment ended[58]. This example highlights the difference between drugs that are effective acutely, and those - like iboga and oxa-iboga - that produce lasting alterations of SUD-related behavioral phenotypes.

There are limitations in the present work that need to be considered and addressed in future studies. In terms of the molecular hypothesis, we propose the differential KOR signaling as a rationale for the lack of adverse effects of oxa-noribogaine such as aversion and depression-like behaviors[59-64]; however, the inherited iboga polypharmacology may also contribute to the in vivo pharmacological profile of oxa-noribogaine[65]. Detailed studies will be required to elucidate the downstream signaling pathways and the mechanisms underlying the amplification of partial agonism effects upstream (partial G protein activation) to full agonism-like effects downstream (e.g. antinociception), in the context of global signaling signatures and circuitry reformatting that may drive the behavioral changes[59]. With respect to the lasting effects of iboga alkaloids on opioid/drug intake (well beyond compound's clearance), the induction of GDNF (and likely other neurotrophic factors) has been the leading molecular explanatory model in the literature[28,29,65,66]. In the present case, we show long-term induction of GDNF protein by oxa-noribogaine, which

is in line with the GDNF hypothesis. However, we have not examined the causal relationship between the GDNF protein levels in the VTA/mPFC and any of the behavioral consequences. We did not explore alternative intervention designs in which, for example, oxa-noriboigaine would be administered during abstinence. It is plausible that induction of GDNF and other neurotrophins in this phase could facilitate a neurobiological incubation process resulting in increased drug responding and reinforcement[35,44]. In our view this is an unlikely scenario, if the growing evidence from ibogaine clinics provides any clues in this direction. Further, it has been reported that GDNF can lead to toxic effects in the cerebellum[67]. We did not examine GDNF levels or toxicity in the cerebellum, but no long-term cerebellum-related motor effects were observed in the examined dose range of oxa-noribogaine.

The KOR-dependent GDNF induction is noteworthy and should be viewed as part of the pharmacological characterization of oxa-noribogaine, while providing an ample rationale for future examination of the neurotrophic hypothesis of the iboga and oxa-iboga compounds.

For the OUD-related behavior studies, only male rats were used in the opioid self-administration, reinstatement, and OIH experiments, while sex differences in KOR-mediated physiological and pharmacological effects have been reported[33]. In mice there were small differences between sexes in a series of behavioral readouts in response to oxa-noribogaine; however female rat subject will need to be examined in the future OUD efficacy-like studies. We also note that some of the acute behaviors can be confounded by the sedative effects of oxa-iboga compounds in rodents. However, the operant food responding provides a better control (than simple sedation assessment) for the opioid self-administration assays in rats at the same doses. Finally, the possibility of an active metabolite was not examined in this study.

While many important questions remain, they should not obscure the key purpose and focus of the current report – the design and discovery of pharmacological agents in the iboga space with a considerable therapeutic potential. In summary, the oxa-iboga alkaloids are a class of compounds that maintain and enhance the ability of iboga compounds to effect lasting alteration of addiction-related behaviors while addressing iboga's cardiac liability.

## Methods

### Ethical considerations

All experimental procedures involving animals were approved by the Columbia University, Memorial Sloan Kettering Cancer Center, Rutgers University or High Point University Institutional Animal Care and Use Committee (IACUC) and adhered to principles described in the National Institutes of Health Guide for the Care and Use of Laboratory Animals and abide by the ARRIVE guidelines.

All human hearts used for the present study were not suitable for transplantation and originated solely from deceased organ donors in the United States (US). Research was carried out by investigators not involved in donor recruitment and all human originating material was received de-identified. As such the study is classified as non-human subjects research (NHR) and does not require prior Institutional Review Board (IRB) approval. Written legal, study nonspecific consent was obtained by the Organ Procurement Organizations (OPOs) from the donor or next of kin wishes. No compensation was provided for the organ donations.

### Statistics & reproducibility

Number of replicates and statistical analysis are reported in the figure and table legends, in the supplementary information and source data files. All comparisons were planned before performing each experiment based on literature reports and/or preliminary experiments. Statistical methods were not used to predetermine sample sizes, with the exception that power analyses were conducted to estimate the number of animal subjects required for in vivo experiments.

For self-administration studies sample size calculations were used to determine the number of subjects required for $\alpha = 0.05$ and $\beta = 0.9$, assuming variance determined from previous studies for specific inferential statistical analyses.

Outlier data were not excluded from analysis. Subjects that did not complete the treatment regimens during self-administration studies were excluded from analysis.

Data were replicated using technical and independent replicates. See figure and table legends, the supplementary information and source data files for specific details.

For self-administration studies pseudo-randomization prior to initiation of self-administration training was used.

In vitro pharmacological characterization, MOR/KOR KO and control WT tail-flick tests, cardiotoxicity study, pharmacokinetic and protein/tissue binding experiments were performed and analyzed by experimenters blinded to the chemical identity of the test substance.

Antagonism tail-flick and open-field experiments were performed unblinded, due to the fact that the specific traits of the antagonists used (pre-administration time, receptor target) needed to be known to correctly perform the experiments.

The overt behavioral effects of the test compounds precluded blinding of drugs or doses during self-administration studies.

### Biological systems

**Cell lines.** HEK293T (CRL-3216) was purchased on from American Type Culture Collection (Rockville, MD, USA). CHO-K1 cells were obtained from the American Type Culture Collection American Type Culture Collection (Rockville, MD, USA). No commonly misidentified cell lines were used in the study. Cells have not been authenticated after purchase nor tested for mycoplasma.

For assays performed by contracted research organizations, cells were procured by the specified service provider.

**Primary human cardiomyocytes (ex vivo human tissue, AnaBios corporation).** AnaBios Corporation's procurement network includes only US based Organ Procurement Organizations (OPOs) and Hospitals. Policies for donor screening and consent are the ones established by the United Network for Organ Sharing (UNOS). Organizations supplying human tissues to AnaBios follow the standards and procedures established by the US Centers for Disease Control (CDC), are inspected biannually by the Department of Health and Human Services (DHHS), are certified by Centers for Medicare & Medicaid Services (CMS) and must abide by CMS regulations. Tissue distribution by OPOs is governed by their internal Institutional Review Board (IRB) procedures in compliance with Health Insurance Portability and Accountability Act (HIPAA) regulations regarding patient privacy. All transfers of donor organs to AnaBios are fully traceable and periodically reviewed by US Federal authorities. Donor characteristics are reported on (Fig. S8F) and exclusion criteria were previously described[68]. Only tissue material originating from adult male donor hearts was used within the study.

**Animals.** All animals received regular veterinary care (weekly by institutional veterinarians) including daily health monitoring (by experimenters) of the animals (observing home cage behaviors, nesting, and body weight). All procedures were designed to minimize any stress/distress.

**Surgical procedures considerations.** Post-surgical monitoring of animals occurred every 20 min from completion of surgery until sternal recumbency (on all four legs); every 60 min until mobile and then daily. For behavioral procedures, the experimenter monitored the subject in the chamber and the computer to ascertain the drug effects for each session. The investigator also removed the subject from the experimental chamber and transferred the subject to the

respective home cage. The experimenter monitored activity, overall appearance, and signs of distress after returning the subject to the home cage. Signs of distress include lack of grooming, food left in the chamber, significant weight loss (>80% of free-feeding weight), and signs of infection (lethargy, vocalization upon handling). Veterinarians were consulted prior to initiating any antibiotics treatment for infection.

**Diet and feeding conditions.** Reverse osmosis water (ultraviolet light treated) and food were provided ad libitum, with no restriction on feeding prior to experiments with exception of lever training and food operant responding (details provided in the corresponding sections).

Columbia University (mice): PicoLab® Rodent Diet #5053 or 5001 (protein ≥20.0/23.0%, fat ≥4.5%, fiber ≤6.0%, ash ≤7.0/8.0% and moisture ≤12.0%).

Rutgers University (mice): irradiated PicoLab® Purina #5058 (protein 20.0%, fat 9.0%, fiber 4.0%, ash ≤6.5% and moisture ≤12.0%).

High Point University (rats and mice): 2018 Tekland Global 18% rodent diet (Envigo), for lever training and food operant responding studies a custom manufactured food pellets were used 45 mg MLab Rodent Tablet 20 mg (TestDiet, Inc., #1811156 (5TUM)).

Sai Life Sciences (rats and mice): T.2014C.15 Teklad global Certified Rodent diet with 14% Protein (Envigo Research Private Ltd, Hyderabad).

**Euthanasia practices.** Columbia University (mice): $CO_2$ asphyxiation with death ensured after euthanasia by a secondary method of euthanasia: decapitation/cervical dislocation or bilateral thoracotomy.

Rutgers University (mice): $CO_2$ asphyxiation using a SmartBox automated $CO_2$ system (euthanex.com) as approved by Rutgers IACUC.

High Point University (rats and mice): Subjects were placed individually into an anesthesia induction chamber (VetEquip; 10 × 9.375 × 6.5 in (L x W x H)) containing an air tight lid, fresh gas inlet connected to a precision extended-capacity isoflurane vaporizer (Patterson Veterinary Tec-3 Vaporizer, #:07-870-3592), and an evacuation outlet connected to an activated charcoal absorption filter trap (VaporGuard, VetEquip; # 931401). $O_2$ was used as the carrier gas. The output was adjusted to 4.5 L/min, with an initial flow rate of 2% gradually increasing to 5% over 2 min. Rats lost consciousness within 3 min and the flow remained until respiration ceased and death ensued (5-6 min). Decapitation was used as the secondary (adjunctive) euthanasia method.

Sai Life Sciences (rats and mice): For tissue collection (PK and tissue homogenate binding) animals were sacrificed under deep Isoflurane anesthesia by cutting open the abdominal vena-cava and perfusing the whole body with saline from heart.

General euthanasia by asphyxiation in a $CO_2$ euthanasia chamber (up to 3 rats or 5 mice), not applicable for tissue collection. The chamber was filled with $CO_2$ (at a flow rate sufficient to fill 30–70% of the chamber volume per minute) until subjects fell unconscious. Once respiratory movement ceased, $CO_2$ flow was maintained for an additional minute and animals were kept in the chamber for two additional minutes after the flow was stopped. Death was confirmed after cessation of life signs (heartbeat, respiration, and pupil dilation) by a secondary method of euthanasia: bilateral thoracotomy or exsanguination.

**Animal subjects used.** Tail-flick, open-field and forced swim tests (Columbia and Rutgers University): Healthy adult C57BL/6 J mice male (8–15 weeks, 29-35 g) and female (8–15 weeks, 19–26 g) were purchased from the Jackson Laboratory (Bar Harbor, ME) and housed 5 mice per cage. Mice were maintained on a 12 h light/dark cycle (lights on 7:00-19:00) and all testing was done in the light cycle. The temperature was kept constant at 22 ± 2 °C, and relative humidity was maintained at 50 ± 5%.

Conditioned placed preference (High Point University): Healthy adult male and female C57BL/6 mice (10–15 weeks, 25–32 g) were purchased from the Jackson Laboratory (Bar Harbor, ME) and housed in groups of 4 to 6 in AllerZone MicroIsolator cages placed in RAIR HD Enviro-Gard™ Ventilated racks with HEPA Filtered Air Units. Vivarium conditions were recorded daily, and the automated climate control ensured constant temperature in the range of 20–24 °C and humidity 40–60%. Mice were maintained on a 12 h light/dark cycle with lights on at 18:00 and experimental sessions took place during the dark phase of the cycle.

Self-administration and neurotropic factors expression studies: Adult male Fisher F-344 rats (CDF Strain, 90–150 days, 230–280 g) were purchased from Charles River Laboratories (Wilmington, MA) and housed in groups of 2 in AllerZone MicroIsolator cages placed in RAIR HD Enviro-Gard™ Ventilated racks with HEPA Filtered Air Units prior to surgery. Following surgery rats were housed individually in acrylic cages. Vivarium conditions were recorded daily, and the automated climate control ensured constant temperature in the range of 20–24 °C and humidity 40-60%. Rats were maintained on a 12 h light/dark cycle with lights on at 18:00, and experimental sessions took place during the dark phase of the cycle.

Pharmacokinetic studies: Healthy adult male C57BL/6 mice (8–12 weeks, 18–36 g) or healthy adult male Wistar rats (8–12 weeks, 250–280 g) were procured from Global (India). Three mice or rats were housed in each cage. Temperature and humidity were maintained at 22 ± 3 °C and 30–70%, respectively and illumination was controlled to give a sequence of 12 h light and 12 h dark cycle. Temperature and humidity were recorded by auto-controlled data logger system.

## In vitro studies

**Broad receptor panel assay.** Binding constants ($K_i$) at the selected human receptors, ion channels and transporters were generously determined using radioligand displacement experiments by the National Institute of Mental Health's Psychoactive Drug Screening Program, Contract # HHSN-271-2008-00025-C (NIMH PDSP)[69]. The NIMH PDSP is Directed by Bryan L. Roth MD, PhD at the University of North Carolina at Chapel Hill and Project Officer Jamie Driscoll at NIMH, Bethesda, MD, USA. For experimental details please refer to the PDSP website http://pdsp.med.unc.edu/.

**Commercial in vitro assays.** KOR [$^{35}$S]GTPγS agonist assay (for noribogaine, item 333200), rat NMDA (Glutamate, [$^3$H]MK-801, item 233010), human NOP (Orphanin ORL1, item 260600) and hERG ([$^3$H] Dofetilide; item 4094) radioligand displacements and human nicotinic acetylcholine receptor (nAChR $\alpha_3\beta_4$, item CYL8057IF2) inhibition studies were performed by contracted research organization. Experiments were conducted as described on https://www.eurofinsdiscoveryservices.com/.

**hSERT inhibition assay.** Stably transfected hSERT-HEK cellular cultures were maintained in Dulbecco's Minimal Essential Medium (DMEM) with GlutaMAX (Gibco-Thermo Fisher Scientific, Waltham, MA) with the following additions: 10% (v/v) Fetal Bovine Serum (FBS, Atlanta Biologicals, Flowery Branch, GA), 100 U/mL Penicillin 10 µg/mL Streptomycin (Gibco-Thermo Fisher Scientific, Waltham, MA), and 500 µg/mL Geneticin (G418) (Sigma Aldrich, St. Louis, MO). Singly transfected cells were seeded at a density of 0.09 × $10^6$ cells/well in poly-D-Lysine (Sigma Aldrich, St. Louis, MO) coated white solid-bottom 96-well plates (Greiner Bio-One, Kremsmünster, Austria). Cells were grown for 44 h in aqueous media at 37 °C under 5% $CO_2$ atmosphere. At the beginning of the experiment, the cellular growth solution was aspirated, and individual cells were rinsed with 150 µL of 1 × Dulbecco's Phosphate Buffered Saline (PBS; HyClone Laboratories, Logan, UT). 63 µL of Experimental Media (DMEM without phenol red but with 4.5 g/L of D-Glucose (Gibco-Thermo Fisher Scientific,

Waltham, MA), 1% (v/v) FBS, 100 U/mL Penicillin, and 10 µg/mL Streptomycin) with tiered concentrations of inhibitor (vehicle: DMSO and control inhibitor: Imipramine (Sigma Aldrich, St. Louis, MO)) were added[70]. After pre-incubation period (one hour), 63 µL of Experimental Media containing various concentrations of tested inhibitor (or vehicle) along with 2 × a specified amount of fluorescent substrate APP+ (final concentration: 1.1 µM, Sigma Aldrich, St. Louis, MO)[71] was added to the wells. After fluorescent probe was allowed to uptake for 30 min, the contents of each well were aspirated and consequently, rinsed twice with 120 µL of PBS. A final solution of 120 µL of PBS was finally added to all corresponding wells for cell maintenance before measuring fluorescence uptake (BioTek H1MF plate reader (BioTek Instruments, Winooski, VT), APP+ excitation and emission wavelengths 436 and 500 nm). Recorded inhibitor values were first subtracted from vehicular values to quantify the respective fluorescence uptake. Data were analyzed using the dose-response-inhibitor nonlinear curve fitting model (log[inhibitor] vs response (four parameters) in Graphpad Prism 10 (Graphpad Software, La Jolla, CA).

**Radioligand competition binding assays (Mouse opioid receptors).** Assays were performed as previously reported[72]. Briefly, IBNtxA and [125I]BNtxA were synthesized at MSKCC as previously described[73–75]. Na125I was purchased from Perkin-Elmer (Waltham, MA). [125I]BNtxA binding was carried out in membranes prepared from Chinese Hamster Ovary (CHO) cells stably expressing murine clones of MOR, DOR, and KOR, as previously described[73,74,76]. Binding incubations were performed at 25 °C for 90 min in 50 mM potassium phosphate buffer, pH 7.4, containing 5 mM magnesium sulfate. After the incubation, the reaction was filtered through glass-fiber filters (Whatman Schleicher & Schuell, Keene, NH) and washed three times with 3 mL of ice-cold 50 mM Tris-HCl (pH 7.4) on a semiautomatic cell harvester. Nonspecific binding was defined by addition of levallorphan (8 µM) to matching samples and was subtracted from total binding to yield specific binding. Protein concentrations were determined using the Lowry method with BSA as the standard[77]. Data were analyzed using the binding-competitive nonlinear curve fitting model (One site - Fit $K_i$) available in Graphpad Prism 10 (Graphpad Software, La Jolla, CA).

**[35S]GTPγS functional assay (mouse opioid receptors).** Assays were performed as previously reported[72]. Briefly, [35S]GTPγS binding was performed on membranes prepared from transfected cells stably expressing opioid receptors in the presence and absence of the indicated compound for 60 min at 30 °C in the assay buffer (50 mM Tris-HCl (pH 7.4) 3 mM MgCl₂, 0.2 mM EGTA, and 10 mM NaCl) containing 0.05 nM [35S]GTPγS; 2 µg/mL each leupeptin, pepstatin, aprotinin, and bestatin, and 30 µM GDP, as previously described[78]. After the incubation, the reaction was filtered through glass fiber filters (Whatman Schleicher & Schuell, Keene, NH) and washed three times with 3 mL of ice-cold buffer (50 mM Tris-HCl, pH 7.4) on a semiautomatic cell harvester. Filters were transferred into vials with 3 mL of Liquiscint (National Diagnostics, Atlanta, GA), and the radioactivity in vials was determined by scintillation spectroscopy in a Tri-Carb 2900TR counter (PerkinElmer Life and Analytical Sciences). Basal binding was determined in the presence of GDP and the absence of drug. Data were normalized to 1000 nM DAMGO, DPDPE, and U50,488 for MOR, DOR, and KOR binding, respectively. Data were analyzed using the dose-response-stimulation nonlinear curve fitting model (log[agonist] vs. response (three parameters)) in Graphpad Prism 10 (Graphpad Software, La Jolla, CA).

**BRET functional assays (G-protein, β-arrestin and Nb33 recruitment).** HEK-293T cells were obtained from the American Type Culture Collection (Rockville, MD) and were cultured in a 5% CO₂ atmosphere at 37 °C in Dulbecco's Modified Eagle Medium (high glucose #11965; Life Technologies Corp.; Grand Island, NY) supplemented with 10%

Fetal Bovine Serum (FBS, #35-010-CV, Corning, Corning, NY, USA), 100 U·mL⁻¹ penicillin (#30-002-CI, Corning, Corning, NY, USA), and 100 µg·mL⁻¹ streptomycin (#30-002-CI; Corning, Corning, NY, USA). The following chemicals were used without further modification: coelenterazine H (#DC-001437, Dalton Pharma Services, Toronto, ON, Canada), PEI (#NC1014320, Polysciences, Warrington, PA, USA), DAMGO (Abcam, Cambridge, United Kingdom) and (±)-U-50488 HCl (Tocris Biosciences, Minneapolis, MN, USA).

DNA Constructs (G-protein, β-arrestin, and Nb33): The rat KOR (rKOR) and mouse MOR (mMOR) were provided by Dr. Lakshmi Devi at Mount Sinai School of Medicine. The human KOR (hKOR) and MOR (hMOR) were obtained from the Missouri S&T Resource Center. The GRK3, Gα_oB with Renilla luciferase 8 (RLuc8) inserted at position 91 (Gα_oB-RLuc8), and Gβ₁ (β₁) were provided by C. Galés[79,80]. Venus-Arrestin2 and Gγ₂, which was fused to the full-length mVenus at its N-terminus via the amino acid linker GSAGT (mVenus-γ2), were constructed in house. The expression vectors coding for rat/human KOR and mouse/human MOR tagged at the C-terminus with Nanoluc (KOR-nluc or MOR-nluc) and were constructed using standard techniques in molecular biology and confirmed by DNA sequencing (Genewiz, South Plainfield, NJ, USA). Briefly, three DNA inserts were PCR amplified, one coding for the N-terminal signal peptide and flag tag, one coding for KOR or MOR, and one coding for nanoluc. The inserts were ligated and cloned into a pcDNA3.1 (+) vector (#V79020, ThermoFisher Scientific, Waltham, MA, USA). The plasmid coding for the nanobody33Venus (Nb33) construct[49] was a gift from Dr. Meritxell Canals at the University of Nottingham.

The following cDNA amounts were transfected into HEK-293T cells (4 × 10⁶ cells/plate) in 10 cm dishes using polyethylenimine (PEI) in a 1.5:1 ratio (diluted in DMEM, Life Technologies). G-protein β-γ release: 2.5 µg KOR, 0.1 µg Gα_oBRLuc8, 6.2 µg β₁, 6.2 µg mVenus-γ2. Arrestin recruitment: 0.2 µg KOR-nluc, 15 µg Arrestin-2-Venus, with/without 5 µg GRK3. Cells were maintained in the HEK-293T media described above. After 24 hours the media was changed, and the experiment was performed 48 h after transfection. Nb33 recruitment: A total of 5 µg of cDNA was transiently transfected into HEK-293T cells (2 × 10⁶ cells per plate) in 10 cm dishes (1 µg receptor-nluc, and 4 µg Nb-33-Venus), using PEI in a 6:1 ratio (diluted in DMEM). Cells were maintained in the HEK-293T media described above. Experiments were performed 48 h after transfection.

Transfected cells were dissociated and re-suspended in phosphate-buffered saline (PBS). Approximately 200,000 cells/well were added to a black-framed, white-well, 96-well plate (#60050; Perkin Elmer; Waltham, MA). At time zero, the luciferase substrate coelenterazine H (5 µM) was added to each well. Ligands were added after 5 min, then BRET signal was measured 5 min later for G-protein, and 10 min later for Nb33 and β-arrestin recruitment. BRET measurements were performed using a PHERAstar FS plate reader (BMG Labtech, Cary, NC, USA). The BRET signal was calculated as the ratio of the light emitted by the mVenus acceptor (510–540 nm) over the light emitted by the NanoLuc donor (475 nm). This drug-induced BRET signal was normalized using the $E_{max}$ of U-50,488 (for KOR) and DAMGO (for MOR) as the maximal response at KOR or MOR. Data were analyzed using the dose-response-stimulation nonlinear curve fitting model (log[agonist] vs. response (four parameters)). All experiments were repeated in three independent trials, each with triplicate determinations.

**TRUPATH BRET based assays.** Assays were performed as previously reported[23,81]. Briefly, cells were plated either in 6-well dishes at a density of 700,000–800,000 cells per well, or 10 cm dishes at 7–8 million cells per dish. Cells were transfected 2–4 h later, using a 1:1:1:1 DNA ratio of receptor: Gα-RLuc8:Gβ:Gγ-GFP2 (100 ng per construct for 6-well dishes, 750 ng per construct for

10 cm dishes), except for the Gγ-GFP2 screen, in which an ethanol coprecipitated mixture of Gβ1–4 was used at twice its normal ratio (1:1:2:1). Transit 2020 (Mirus Biosciences) was used to complex the DNA at a ratio of 3 μl Transit per μg DNA, in OptiMEM (Gibco-ThermoFisher) at a concentration of 10 ng DNA per μl OptiMEM. The next day, cells were harvested from the plate using Versene (0.1 M PBS + 0.5 mM EDTA, pH 7.4) and plated in polyD-lysine-coated white, clear-bottom 96-well assay plates (Greiner Bio-One) at a density of 30,000–50,000 cells per well. One day after plating in 96-well assay plates, white backings (PerkinElmer) were applied to the plate bottoms, and growth medium was carefully aspirated and replaced immediately with 60 μl of assay buffer (1× Hank's balanced salt solution (HBSS) + 20 mM HEPES, pH 7.4), followed by a 10 μl addition of freshly prepared 50 μM coelenterazine 400a (Nanolight Technologies). After a 5 min equilibration period, cells were treated with 30 μl of drug for an additional 5 min. Plates were then read in an LB940 Mithras plate reader (Berthold Technologies) with 395 nm (RLuc8-coelenterazine 400a) and 510 nm (GFP2) emission filters, at integration times of 1 s per well. Plates were read serially six times, and measurements from the sixth read were used in all analyses. BRET2 ratios were computed as the ratio of the GFP2 emission to RLuc8 emission. Data were analyzed using the dose-response-stimulation nonlinear curve fitting model (log[agonist] vs. response (three parameters)). All experiments were repeated in three independent trials each with duplicate determinations.

**Cardiotoxicity assay in adult human primary cardiomyocytes (AnaBios corporation).** Cardiotoxicity of noribogaine and oxa-iboga analogs was assessed according to published procedure[40,41]. Briefly, adult human primary ventricular myocytes were isolated from ethically consented donor hearts that were enzymatically digested using a proprietary protocol. Cardiomyocytes were placed in a perfusion chamber mounted on the stage of inverted Motic AE31E (IonOptix) or Olympus IX83P1ZF (MyoBLAZER) microscope and continuously perfused at approximately 2 mL/min with recording buffer heated to 35 ± 1 °C using an in-line heater from Warner Instruments (IonOptix & MyoBLAZER) and allowed to equilibrate for 5 min under constant perfusion. The cells were field stimulated with supra-threshold voltage at a 1 Hz pacing frequency, with a bipolar pulse of 3 ms duration, using a pair of platinum wires placed on opposite sides of the chamber connected to a MyoPacer stimulator. Starting at 1 V, the amplitude of the stimulating pulse was increased until the cardiomyocytes started generating contractility transients, and a value 1.5× threshold was used throughout the experiment. Cardiomyocytes were then imaged at 240 Hz using an IonOptix MyoCam-S CCD camera (IonOptix) or at 148 Hz using an Optronis CP70-16-M/C-148 (MyoBLAZER) camera. Digitized images were displayed within the IonWizard acquisition software (IonOptix) or Myo-BLAZER acquisition software. The longitudinal axis of the selected cardiomyocyte was aligned parallel to the video raster line, by means of a cell framing adapter. Optical intensity data was collected from a user-defined rectangular region placed over the cardiomyocyte image. The optical intensity data represented the bright and dark bands corresponding to the Z-lines of the cardiomyocyte. The IonWizard software or MyoBLAZER Analysis software analyzed the periodicity in the optical density of these bands by means of a fast Fourier transform algorithm.

Compound test solutions were formulated from stock solutions within 30 min prior to experimental application to the cells. Test solutions were applied after vehicle control (120 s interval, 1 Hz stimulation) in an increasing concentration order (in 300 s intervals, 1 Hz stimulation) and experiment was terminated after wash control (300 s interval, 1 Hz stimulation).

Positive control 30 nM ATX-II (toxin from anemonia sulcate) was applied after vehicle control (120 s interval, 1 Hz stimulation) and the data were recorded (300 s interval, 1 Hz stimulation).

An aftercontraction (AC) was visually identified as spontaneous secondary change in the slope of the contractility transient that occurred before the next stimulus-induced contraction and that produced an abnormal and unsynchronized contraction. Contraction Failure (CF) was also visually identified when an electrical stimulus was unable to induce a contraction. Alternans and Short-Term Variability (STV) are visualized in Poincaré plots of Contraction Amplitude variability. STV (STV = $\Sigma|CA_{n+1} - CA_n| (20 \times \sqrt{2})^{-1}$) was calculated with the last 20 transients of each control and test article concentration period ($CA_n$ – contractility amplitude, nth in sequence). Alternans were identified as repetitive alternating short and long contractility amplitude transients. STV values were normalized to the vehicle control value of each cell. AC, CF and Alternans were plotted and expressed as % of incidence of cells exhibiting each of the signals normalized by the total number of cardiomyocytes.

The concentration range was selected on the basis of these considerations: the free plasma noribogaine $C_{max}$ values were estimated from the reported clinical data of total plasma concentrations[39] and our own human plasma protein binding data (Fig. S7g); approximately 100 nM noribogaine (60 mg oral dose) produced a mild QT effect and represents a relatively safe plasma level, while 300–400 nM (180 mg oral dose) induced concerning levels of QT prolongation ( > 500 ms), whereas low micromolar concentrations can be reached after detox therapeutic doses of ibogaine ( > 8 mg/kg) which have been associated with risk of arrhythmias.

## In vivo pharmacology and behavioral experiments

**Test compounds and formulations.** Drugs for in vivo experiments were synthesized in the Sames laboratory as described in the synthesis section (noribogaine hydrochloride and oxa-iboga analogs) or obtained from following sources and used as received: U50,488 hydrochloride (Tocris Bioscience), cyprodime hydrochloride (Tocris Bioscience), naloxone hydrochloride (Alfa Aesar), naltrindole hydrochloride (MedChemExpress), imipramine hydrochloride (Alfa Aesar), aticaprant free base (MedChemExpress), morphine sulfate (Gallipot, Inc., St. Paul, MN), penicillin G procaine (Butler Company, Columbus, OH), propofol (Patterson Veterinary Supply, Inc., Loveland, CO), fentanyl citrate (Fagron, Inc., St Paul, MN), ketamine hydrochloride (Ketaset), xylazine (Xylamed, Patterson Veterinary Supply, Inc., Greely, CO), cocaine hydrochloride and heroin hydrochloride (Drug Supply Program of the National Institute on Drug Abuse, Bethesda, MD). The compound solutions for pharmacology experiments (tail-flick, place conditioning preference, open field and forced swim test) were prepared on the same day as testing from pure solid material. Solids were dissolved in UPS grade 0.85% saline with addition of 2 molar equivalents of glacial acetic acid. Heat and sonication were used to assist in fully dissolving the solid to obtain a clear solution. Aticaprant was solubilized using 7% (v/v) Tween 80 in saline with addition of 2 molar equivalents of glacial acetic acid. Compound solutions were prepared at a concentration allowing for a volume of injection (190–350 μL) based on the body weight of the animal.

For self-administration and GDNF expression experiments test compounds (noribogaine, oxa-noribogaine and epi-oxa-noribogaine) were dissolved in vehicle (2 molar equivalents of acetic acid in water) at a concentration of (20 mg/mL) to allow for injection volume of (2 mL/kg). For example, a 40 mg/kg dose administered to a 300 g rat required injecting 12 mg of test substance in 0.6 mL of vehicle (0.5 mg/kg/kg). Heat and sonication were used to assist in fully dissolving the solid to obtain a clear solution.

For pharmacokinetic study, noribogaine hydrochloride solution for subcutaneous (s.c.) administration was prepared by dissolving the solid material in DMSO and the stock solution was diluted with normal

saline to obtain a final formulation in 5% DMSO, 95% (v/v) normal saline (formulation strength 2 mg/mL). The final mixture was vortexed and sonicated to obtain a clear solution. Oxa-noribogaine was solubilized using 0.85% saline with addition of 2 molar equivalents of glacial acetic acid, formulations were vortexed and sonicated to obtain a clear solution (formulation strength 2 mg/mL for s.c. and 4 mg/mL for i.p. administration).

**Tail-flick test.** Tests were performed as previously reported[24]. Mice were moved to the testing room 30 min before the experiment to allow for acclimation. The body weight of each mouse and base tail-flick value were recorded. Mice were administered a 1 mg/kg s.c. dose of compound solution. After injection, mice were returned to the home cage and allowed to rest for 30 min. Thirty minutes post injection, the tail-flick measurement was taken using thermal stimulation via IR on a Ugo Basile unit set to 52 PSU (to achieve a baseline between 2 and 3 s and 10 s was used as a maximum latency to prevent tissue damage). Mice were then administered 3 mg/kg s.c. dose, allowed to rest for 30 min, followed by another tail-flick measurement. This process was repeated for doses 10 and 30 mg/kg in increasing order. Tail-flick latencies for the different doses were expressed as percentage of maximum potential effect (% MPE) by subtracting the experimental value by the base tail flick value then dividing by the difference between the maximum possible latency (10 s) and the base tail-flick value and finally multiplying by 100.

ED$_{50}$ value for U50,488 was determined using four separate groups of naive mice (5 animals for each dose, baseline latency was determined for each animal prior to drug administration), as cumulative dosing of U50,488 in the same animal resulted in a rapid development of tolerance.

ED$_{50}$ values were calculated using the dose-response-stimulation nonlinear curve fitting model (log[agonist] vs. response (four parameters)) with applied constraints Top = 100 and Bottom = 0.

**Tail-flick test (KO animals).** For analgesic testing in knockout animals MOR-1 Exon-1 KO[82] and KOR-1 Exon-3 KO[83] mice on a C57 background were bred in the Pintar laboratory at Rutgers University. All mice used were opioid naïve. Analgesia was tested in wild-type and KO animals by the radiant heat tail-flick technique using an IITC Model 33 Tail Flick Analgesia Meter as previously described[84,85]. The intensity was set to achieve a baseline between 2 and 3 s. Tail flick antinociception was assessed as an increase in baseline latency, with a maximal 10 s latency to minimize damage to the tail. Data were analyzed as percent maximal effect, % MPE, which was calculated according to the formula: % MPE [(observed latency − baseline latency)/(maximal latency − baseline latency)] × 100. Compounds were administered subcutaneously (s.c.) and analgesia was assessed at the peak effect (15 min). Mice were tested for analgesia with cumulative subcutaneous doses of the drug until the mouse can withstand the maximal latency. Once the mouse reached the maximal latency, the mouse was no longer given higher doses.

ED$_{50}$ values were calculated using the dose-response-stimulation nonlinear curve fitting model (log[agonist] vs. response (four parameters)) with applied constraints Top = 100 and Bottom = 0. Statistical significance (WT vs KO-model) was assessed using the extra-sum-of-squares F test (alpha = 0.05).

**Tail-flick test (antagonism pre-treatment).** Mice were transferred to the testing room 30 min before the experiment to allow for acclimation. The weight of each mouse and base tail-flick value were recorded. Mice were administered the selected antagonist s.c. Antagonists were prepared and used at the following doses: cyprodime 10 mg/kg, naloxone 10 mg/kg, naloxone 1 mg/kg, naltrindole 0.5 mg/kg, and aticaprant 0.1 mg/kg. After injection, mice were returned to the home cage and allowed to rest for a duration of time specific to the

antagonist being used. The wait time after administration of antagonist are as follows: cyprodime 30 min, naloxone 30 min, naltrindole 45 min and aticaprant 20 min. Mice were then administered oxa-noribogaine s.c. prepared at the dose of 10 mg/kg. After injection, mice were returned to the home cage to rest for a duration of time specific to the experimental compound being used. The wait time after administration of oxa-noribogaine was 30 min. Afterwards, the tail-flick measurement was taken as described above.

Statistical analysis was performed using the unpaired t-test, two-tailed with Welch's correction.

**Open field test.** Mice were transferred to the testing room 30 min before the experiment to allow for acclimation. Body weight of each mouse was recorded. For experiments with antagonist, pre-treatment, the dosing and after application wait times were the same as for the tail-flick test. Mice were administered an ED$_{50}$, ED$_{80}$ or >ED$_{95}$ doses of test compound (volume of injection 290–350 μL based on body weight). After injection, mice were returned to the home cage and allowed to rest for 30 min. Each mouse was then gently placed in the center of a clear Plexiglas arena (27.31 × 27.31 × 20.32 cm, Med Associates ENV-510) lit with dim light ( ~ 5 lux) and allowed to ambulate freely for 60 minutes. The locomotion of the mouse was tracked by infrared beams embedded along the X, Y, Z axes of the area and automatically recorded. Data was collected on Activity Monitor by Med Associates.

Statistical analysis was performed using the unpaired t-test, two-tailed with Welch's correction to analyze total locomotion and a two-way ANOVA with Šidák's multiple comparison test for post hoc comparison to analyze locomotion across time bins.

**Forced swim test.** One week after arriving in the facility, mice were handled for approximately 5 min by a male experimenter. All experiments were carried out by the same male experimenter who performed the handling. On the day of FST testing, mice were weighed and transferred to the testing room 30 min prior to experimentation. Mice were administered 0.85% saline (s.c.), imipramine 15 mg/kg (i.p.), oxa-noribogaine ED$_{80}$ (5.4 mg/kg) or >ED$_{95}$ 10 mg/kg dose s.c. 15 min after receiving saline or imipramine injection, or 30 min after receiving oxa-noribogaine injection, each subject mouse was gently placed into cylinder (plexiglass 40 cm tall, diameter 22 cm) filled with water (height 18 cm) kept at 23.8–25.0 °C. Mice were allowed to swim for 6 min after which they were removed, dried, and returned to their home cage. Video footage was analyzed with Noldus FST scoring software to obtain immobility scores. The mouse's view of the FST cylinder was obstructed prior to initiating the experiment. Only the last 4 min of the 6 min test were analyzed for time spent immobile. All FSTs were performed between the hours of 12:00 and 18:00.

Statistical analysis was performed using the unpaired t-test, two-tailed with Welch's correction.

**Place conditioning preference.** Place conditioning was conducted in a three-chamber apparatus with two equal sized chambers (16.8 × 12.7 × 12.7 cm) that differed in color (black, white) and flooring (mesh, bar), separated by a middle chamber (7.2 × 12.7 × 12.7 cm) with computer-controlled doors and equipped with infrared diodes (Med Associates). Mice were acclimated to the room for three days before the pretest phase.

For determination of pre-conditioning baseline preferences, mice were placed in the middle chamber; the doors were raised, and the animal had free access to the chamber for 20 min. Dependent measures included the amount of time spent in each compartment, activity in the compartments, and entries into the compartments. No statistically significant difference was observed in the amount of time spent between the compartments during the pretest phase. Using an unbiased approach, the drug-paired chamber was randomly assigned

for each mouse. Mice spending more than 65% time in either compartment or 33% in the middle compartment during the pre-conditioning session were excluded from the study. Conditioning began the day following the pretest phase and consisted of three two-day pairing cycles. On the first, third and fifth days of conditioning, mice received either cocaine (10 mg/kg, i.p.), morphine (20 mg/kg, i.p.), oxa-noribogaine (10 or 40 mg/kg, i.p.) or epi-oxa-noribogaine (10 and 40 mg/kg, i.p.) and were confined to the drug paired compartment for 30 min. On the second, fourth, and sixth days, mice were administered saline (1 mL/kg, i.p.) and confined to the vehicle-paired compartment for 30 min.

For testing on the seventh day, mice were placed in the middle compartment, doors were raised, and the mice were allowed access to the entire chamber for 15 min. Dependent measures were identical to the pretest phase.

A CPP score was calculated as the difference in time spent in the drug-paired minus the vehicle paired compartment for both the pretest and testing phases. Data were analyzed using paired t-tests or two-way ANOVA with Šidák's multiple comparison test for post hoc comparison.

**General procedures for intravenous drug self-administration and food responding studies in rats.** Rats were transferred to operant conditioning chambers (ENV-008CT; Med Associates, St. Albans, VT) enclosed in sound-attenuating cubicles (ENV-018; Med Associates). The front panel of the operant chambers contained two response levers (4 cm above the floor and 3 cm from the side walls), a cue light (3 cm above the lever) and a food chute centered on the front wall (2 cm above the floor) that was connected to a food pellet dispenser (ENV-023; Med Associates) located behind the front wall and a tone generator to mask extraneous noise. A syringe pump (PHM-100; Med Associates) holding a 20 mL syringe delivered infusions. A counter-balanced arm containing the single channel liquid swivel was located 8–8.5 cm above the chamber and attached to the outside of the front panel. An IBM compatible computer was used for session programming and data collection (Med Associates Inc., East Fairfield, VT).

For lever training, rats were transferred to operant chambers for daily experimental sessions. Food access was restricted in the home cage, such that rats were maintained at 90% of their normal body weight until responding was engendered at which point food was provided ad libitum. Responding was engendered by delivery of food pellets (45 mg pellets, TestDiet, Richmond, IN) under a fixed ratio (FR 1) schedule of reinforcement. The lever lights were illuminated when the schedule was in effect. Completion of the response requirement extinguished lights, delivered food, and was followed by a 20 s timeout (TO) period during which all lights were extinguished, and responses had no scheduled consequences. After the TO, the lights were illuminated, and the FR schedule was again in effect. Sessions lasted 20 min or until 50 food pellets were delivered.

After operant responding was acquired and maintained by food, subjects were surgically implanted with an intravenous jugular catheter. Venous catheters were inserted into the right jugular vein following administration of ketamine (90 mg/kg; i.p.) and xylazine (5 mg/kg; i.p.) for anesthesia as described previously[86–88]. Catheters were anchored to muscle near the point of entry into the vein. The distal end of the catheter was guided subcutaneously to exit above the scapulae through a Teflon shoulder harness. The harness provided a point of attachment for a spring leash connected to a single-channel fluid swivel at the opposing end. The catheter was threaded through the leash and attached to the swivel. The other end of the swivel was connected to a syringe (for saline and drug delivery) mounted on a syringe pump. Rats were administered penicillin G procaine (75,000 units in 0.25 mL, i.m.) and allowed a minimum of 5 days to recover before self-administration studies were initiated. Catheter patency was maintained by hourly infusions of 0.2 ml of 0.9% saline (w/v) with

1.7 U/ml of heparin equivalent to 0.34 U heparin/infusion while in the home cage only. Infusions of propofol (6 mg/kg; i.v.) were manually administered as needed to assess catheter patency.

Following recovery, rats were transferred to their respective operant chambers for daily self-administration sessions. Before each session, the swivel and catheter were flushed with 500 μL of heparinized saline before connecting the catheter to the syringe mounted on the syringe pump outside of the sound attenuating chamber via a 20 ga luer hub and 28 ga male connector. At the start of each self-administration session, the stimulus light above the active lever was illuminated and both the active and inactive levers were extended. A response on the active lever (FR1) resulted in a 20 s time out (FR1:TO 20 s) during which time the subject received a 200 μL intravenous drug infusion (over the first six seconds), lever light extinguished, levers are retracted, tone is generated, and the house light was illuminated. At the end of the TO, the levers were extended, lever light illuminated, tone silenced, and the house light extinguished. The self-administration session continued for 2 h or until 70 infusions were delivered. Self-administration studies were initiated once stable responding was achieved (defined as total number of infusions ± <20% of the mean of the three previous sessions)[88–90].

**Effect of oxa-noribogaine on morphine self-administration.** Separate groups of rats were trained to self-administer morphine (10, 20 or 40 μg/infusion) under an FR1 schedule of reinforcement. After stable responding was established, rats were administered vehicle (2 mL/kg, i.p.) 30 min prior to the subsequent experimental session. Four days following vehicle administration, rats were administered oxa-noribogaine in an ascending dose manner (3, 10 or 30 mg/kg, i.p.), administered 30 min prior to the beginning of the session. Responding was assessed for seven days following each dose. For each morphine dose, baseline and vehicle active and inactive lever responding was compared using paired t-tests, two tailed. The acute effects of oxa-noribogaine on responding (active and inactive lever) maintained by each morphine dose was assessed independently using a two-way ANOVA with Tukey's multiple comparison test. The effects of oxa-noribogaine on morphine self-administration on active and inactive lever responding for the seven days following each dose of oxa-noribogaine were assessed using a mixed-model analysis of repeated measures data (due to missing data points) with Šidák's multiple comparison test for post hoc comparison.

**Effect of oxa-noribogaine on cue-induced reinstatement of responding previously maintained by morphine or fentanyl.** Two groups of rats were trained to self-administer morphine (20 μg/infusion) or fentanyl (0.625 μg/infusion) in 2 h sessions under FR1 schedule of reinforcement. After a minimum of two weeks of self-administration and three days of stable responding (defined as total number of infusions ± <20% of the mean of the three previous sessions)[88–90] immediately prior to the extinction session, rats underwent operant extinction training in 2 h session in the absence of the drugs and associated cues until responding was <20% of baseline responding maintained by intravenous self-administration of the drug. During the extinction phase, levers were available, but responding on the active lever did not result in cue light presentation or drug delivery.

The day following the last extinction session, each group was divided into two subgroups to receive either vehicle (2 mL/kg, i.p.) or oxa-noribogaine (10 mg/kg, i.p.) 30 min prior to the 60 min session to test for cue-induced reinstatement of morphine or fentanyl. During the assessment, cues previously associated with morphine or fentanyl self-administration were presented following each response on the previously active lever. Responses on the previously inactive lever were recorded but did not result in stimulus presentation.

Baseline responding (three days prior to extinction) on the active and inactive levers were compared separately for the morphine and

fentanyl groups receiving either vehicle or oxa-noribogaine using a two-way ANOVA with time as the repeated measure with Šidák's multiple comparison test for post hoc comparison. For the cue-induced reinstatement test session, responding on the active and inactive levers were analyzed for the morphine and fentanyl groups separately receiving either vehicle or oxa-noribogaine using an unpaired t-test.

**Effect of acute administration of oxa-noribogaine, epi-oxa-noribogaine and noribogaine on morphine self-administration.** Rats were trained to self-administer morphine (10 μg/infusion) as described in the general self-administration training. After stable responding was established, rats were administered vehicle (2 mL/kg, i.p.) 15 min prior to the subsequent experimental session. Three days following vehicle administration, rats were administered oxa-noribogaine (10 or 40 mg/kg, i.p.), epi-oxa-noribogaine (40 mg/kg, i.p.) or noribogaine (40 mg/kg, i.p.), administered 15 min prior to the beginning of the session.

**Effect of acute oxa-noribogaine administration on fentanyl self-administration.** Rats were trained to self-administer fentanyl (625 ng/infusion) as described in the general self-administration training. After stable responding was established, rats were administered vehicle 15 min prior to the subsequent experimental session. Three days following vehicle administration, rats were administered oxa-noribogaine (40 mg/kg, i.p.) administered 30 min prior to the beginning of the session.

**Food maintained responding.** For food maintained responding studies, food access was restricted such that rats were maintained at 90% of their normal body weight until responding stabilized, at which point rats were provided 12–15 g of rodent chow in addition to the food pellets earned during the experimental session.

Rats were lever trained as described in the general methods. Responding was maintained as described with notable difference in timeout (TO) period 6 min and sessions lasting 2 h or until a maximum of 20 pellets was delivered. Once responding stabilized, subjects were assigned randomly to receive oxa-noribogaine (10 and 40 mg/kg, i.p.), epi-oxa-noribogaine (40 mg/kg, i.p.), noribogaine (10 and 40 mg/kg, i.p.) or vehicle administered intraperitoneal 15 min prior to the initiation of the session. Responding was considered stable when variation in the number of reinforcers for three consecutive sessions was less than 20%. For the group receiving oxa-noribogaine, data were analyzed using a two-way ANOVA with Šidák's multiple comparisons test for post hoc comparison.

**GDNF expression studies.** Male Fisher F344 rats were assigned to groups and administered oxa-noribogaine (40 mg/kg; i.p.) or noribogaine (40 mg/kg; i.p.). One day or five days after drug administration, rats from these groups as well as non-treated controls were decapitated, brains removed and placed in a stainless-steel rat brain matrix. Coronal slices were taken and the ventral tegmental area (VTA), nucleus accumbens (NAc) and medial prefrontal cortex (mPFC) were dissected and immediately frozen on dry ice. Total protein was isolated from pulverized tissue (8–15 mg tissue per region was harvested from one subject) from these regions. GDNF contents were assayed using the BiosSensis GDNF, Rat, Rapid™ ELISA assay (Biosensis Pty Ltd, SA, Australia). Protease (Thermo Scientific, Rockford, IL), and phosphatase inhibitors (Cocktails 1 and 2, Sigma-Aldrich, St. Louis, MO) were added to the extraction buffers. Regardless of sample weight 230uL of RIPA buffer (Sigma; R0278) was added into each sample along with protease inhibitors. Samples were sonicated to lyse the tissue (horn sonicator or Diagenode Bioruptor) twice for 15–20 s and incubated on ice for 30 s between sonication. After resting on ice for 30 min, samples were centrifuged at 8,000 RPM (equivalent to 6010 x $g$) for 15 min at 4 °C and the supernatant (total protein lysate)

was transferred to a new tube. Next 8.5 μL of 1 N HCl solution was added to each sample to obtain a pH of 3.5. After 15 min 1.6 μL of 1 N NaOH solution was added to raise the pH to 7.0–7.4. (note: it is important to check pH during acid/base addition steps and ideally practice on blank tissue samples to ensure consistent pH change and measurements; it is critical that pH does not decrease below 3.0 - the risk of permanently denaturing the target). Protein concentrations were measured using the bicinchoninic acid protein assay kit (Pierce, Rockford, IL, USA) on a spectrophotometer (iD5, Molecular Devices, Sunnyvale, CA). Aliquots of 100 μL of isolated protein from each region were transferred to a 96-well ELISA plates. The abundance of GDNF was normalized to the amount of total protein (pg/mg protein). In a separate cohort, we assessed the ability of the kappa antagonist aticaprant to attenuate the oxa-noribogaine induced increase in GDNF in the VTA and mPFC five days after administration. Rats were administered either vehicle (7% (v/v) Tween 80, 2 molar equivalents glacial acetic acid in physiologic saline) or aticaprant (1 mg/kg; s.c.) followed twenty minutes later by administration of either vehicle (2 molar equivalents glacial acetic acid in physiologic saline) or oxa-noribogaine (40 mg/kg; i.p.). Along with non-treated controls, rats were decapitated, and tissue processed as described above. Data were analyzed by protein and region using one way ANOVA with Tukey's multiple comparison test for post hoc comparison.

**Opiate induced hyperalgesia (OIH) study.** Rats were randomly divided into two groups to receive either two morphine (MOR; 75 mg each) or two placebo pellets. Prior to pellet implantation, baseline measurements of paw withdrawal threshold (PWT) using Von Frey (VF) assay were taken. Sustained morphine administration was accomplished by subcutaneous implantation of two morphine pellets (75 mg each). Under light isoflurane anesthesia, an area approximately one inch above the pelvic bone was shaved, a small incision made, two morphine or two placebo pellets were implanted, and the wound was closed by suture[91,92]. Steady state plasma levels of morphine were reported to be achieved 3 days post-pellet implantation and maintained through 12 days post-implantation[91,92].

On Day 8 post-implantation, morphine and placebo pellets were removed under light isoflurane anesthesia and the wound was washed with sterile saline to remove residual morphine. Two hours following pellet removal rats were administered either vehicle (2 molar equivalents acetic acid in saline) or oxa-noribogaine solution (30 mg/kg; I.P.) to obtain four separate study groups.

Paw withdrawal thresholds were recorded for each group in selected time intervals: 8 h, 24 h and 168 h post-implant removal (6, 22 and 166 h post oxa-noribogaine treatment respectively). The presence and development of nociceptive tolerance was assessed by two-way ANOVA with Tukey's multiple comparison test for post hoc comparison.

Electronic Von Frey Assessments: Mechanical sensitivity was measured using the electronic Von Frey 5 with embedded camera (BIO-EVF5; Bioseb US, Pinellas Park, FL). Rats were placed in a modular holder cage (BIO-PVF, Bioseb US) on a wire mesh elevated stand and allowed to acclimate for 5 min prior to assessment. The spring tip of the device was applied to the plantar surface of the hind paw. Brisk paw withdrawal was considered as a positive response, the force (grams) required to initiate paw withdrawal was recorded by the EVF software. The average of a minimum of Two replicates per hind paw was calculated for each rat. The average of all values was calculated as the paw withdrawal threshold (PWT).

**Pharmacokinetic studies (PK, Sai Life Sciences Limited).** Total of twenty-four male mice (twenty-seven male rats) were used per study (3 animals per each time point). Mice were administered subcutaneously (10 mg/kg dose, s.c.), rats intraperitoneally (40 mg/kg dose, i.p.). The

dosing volume for subcutaneous administration was 5 mL/kg and 10 mL/kg for intraperitoneal administration.

Blood samples (~60 μL from mice, ~120 from rats) were collected under light isoflurane anesthesia (Surgivet®) from retro orbital plexus from a set of three animals at specified time points into labeled micro-tubes, containing $K_2EDTA$ solution (20% $K_2EDTA$ solution) as an anticoagulant. Immediately after blood collection, plasma was harvested by centrifugation at 4000 rpm (2000 x g), 10 min at 40 °C and samples were stored at −70 ± 10 °C until bioanalysis. Following blood collection, animals were immediately sacrificed, the abdominal vena-cava was cut open and whole body was perfused from heart using (10 mL for mice or 20 mL for rats) of normal saline. Brain samples were collected from a set of three animals at specified time points. After isolation, brain samples were rinsed three times in ice cold normal saline (for 5–10 s/rinse using (~5–10 mL for mice or ~10–20 mL) of normal saline in disposable petri dish for each rinse) and dried on blotting paper. Brain samples were homogenized using ice-cold phosphate buffer saline (pH − 7.4). Total homogenate volume was three times the tissue weight. All homogenates were stored below −70 ± 10 °C until bioanalysis. The extraction procedure for plasma and brain samples and the spiked plasma and brain calibration standards were identical. A 25 μL of study sample or spiked plasma calibration standard was added to individual pre-labeled micro-centrifuge tubes followed by 100 μL of internal standard prepared in acetonitrile (Glipizide, 500 ng/mL) was added except for blank, where 100 μL of acetonitrile was added. Samples were vortexed for 5 min. Samples were centrifuged for 10 min at a speed of 4000 rpm (~2000 x g) at 4 °C. Following centrifugation, 100 μL of clear supernatant was transferred in 96 well plates and the concentrations of analyte were determined by fit for purpose LC-MS/MS method. Instrument (Waters ACQUITYTM, UPLC, Canada) equipped with (API-4000 MDS Sciex, Applied Biosystems, Canada) detector.

Non-Compartmental-Analysis tool of Phoenix WinNonlin® (Version 8.0 for oxa-noribogaine, Version 7.0 for noribogaine) was used to assess the pharmacokinetic parameters. Peak plasma concentration ($C_{max}$) and time for the peak plasma concentration ($T_{max}$) were the observed values. The areas under the concentration time curve ($AUC_{last}$ and $AUC_{inf}$) were calculated by linear trapezoidal rule. The terminal elimination rate constant ($k_{el}$) was determined by regression analysis of the linear terminal portion of the log plasma concentration-time curve. The terminal half-life ($T_{1/2z}$) was estimated by $0.693/k_{el}$. Clearance (CL) was estimated as $Dose/AUC_{inf}$ and steady state volume of distribution ($V_{ss}$) as CL × MRT (Mean Residence Time). Tissue-to-plasma ratio (Kps) were calculated using Microsoft Excel.

**Plasma protein and brain tissue binding (Sai Life Sciences Limited).** Noribogaine, oxa- and epi-oxa-noribogaine plasma protein and brain tissue binding were determined using rapid equilibrium dialysis (RED) followed by LC-MS/MS quantification in MRM mode. Brain tissue homogenate samples were prepared by diluting one volume of whole brain tissue with three volumes of dialysis buffer (phosphate buffered saline pH 7.4 with 0.1 M sodium phosphate and 0.15 M sodium chloride) to yield 4 times diluted homogenate. A 1 mM stock solution of test compounds was prepared in DMSO and diluted 200-folds in rat plasma (male Sprague Dawley, $n = 5$), human plasma (drug free volunteers; $n = 6$) or rat brain homogenate to prepare a concentration of 5 μM. The final DMSO concentration was 0.5%. Rapid equilibrium dialysis was performed with RED device containing dialysis membrane with a molecular weight cut-off of 8000 Daltons (Thermo Scientific). A 200 μL aliquot of positive control 5 μM and test compound at 5 μM (triplicates) were separately added to the plasma/brain homogenate chamber and 350 μL of phosphate buffer saline (pH 7.4, Thermo Scientific) was added to the buffer chamber of the inserts. After sealing

the RED device with an adhesive film, dialysis was performed in incubator at 37 °C with shaking at 100 RPM for 4 h.

A 50 μL aliquot of positive control and test compounds were added to four 0.5 mL microcentrifuge tubes. Two aliquots were frozen immediately (0 min sample). The other two aliquots were incubated at 37 °C for 4 h along with the RED device.

Following dialysis, an aliquot of 50 μL was removed from each well (both plasma or brain homogenate and buffer side) and diluted with equal volume of opposite matrix (dialyzed with the other matrix) to nullify the matrix effect. Similarly, 50 μL of buffer was added to recovery and stability samples. An aliquot of 100 μL was submitted for LC-MS/MS analysis. Instrument (Waters ACQUITYTM, UPLC, Canada) equipped with (API-4000 MDS Sciex, Applied Biosystems, Canada) detector.

A 25 μL aliquot of positive control and test compounds were crashed with 100 μL of acetonitrile containing internal standard and vortexed for 5 min. The samples were centrifuged at 4000 rpm (~2000 x g) at 4 °C for 10 min and 100 μL of supernatant was submitted for LC-MS/MS analysis in MRM mode. The samples were run without calibration curve and the peak area ratios (analyte versus internal standard) obtained were used to determine the fraction of compound bound to plasma and brain proteins.

**Estimated free drug concentrations, target engagement, and correlation with in vitro pharmacology.** Oxa-noribogaine exhibits rapid and high brain penetration (brain/plasma = 5.5, mice) after subcutaneous administration (s.c.). Approximately 85% of the compound is cleared in the first 2 h and only traces remain after 8 h. We estimated the free plasma and brain drug concentrations based on plasma protein and brain tissue binding ($C_{max(brain)}$ = 166 nM, 10 mg/kg, s.c.). Assuming linearity of PK below 10 mg/kg, $C_{max(brain)}$ at an $ED_{50}$ analgesic dose (3 mg/kg) is expected to be ~50 nM, which matches well the in vitro KOR activation potency ($EC_{50}$ = 41 nM in BRET assay, $EC_{50}$ = 49 nM, in [$^{35}$S]GTPgS assay). Thus, the pharmacodynamic readout (analgesia) correlates with the estimated free drug PK measures. Further, the $C_{max}$ and area under the curve (AUC, drug exposure) values for estimated free drugs are much smaller for oxa-noribogaine in comparison to noribogaine, which is relevant for interpreting the oxa-noribogaine's superior efficacy in addiction models.

**Ibogaine analogs docking studies on KOR.** The active state KOR X-ray crystal structures corresponding to PDB accession code 6B73[93] was extracted from the RCSB server. Three conserved crystallographic waters close of $H^{6.52}$ and $Y^{3.33}$, elucidated in the high-resolution active state MOR X-ray crystal structure (PDB 5C1M)[94] were transplanted to the active state KOR model and their steric positions inside the active state KOR model were optimized by minimization protocols. All the objects except the receptor protein subunit, the crystallized ligand, and the three transplanted waters were deleted, and this was followed by the addition of hydrogens and optimization of the side-chain residues. Ligands were sketched, assigned formal charges, and energy-optimized prior to molecular docking. The ligand docking box for potential grid docking was defined to contain the extracellular half of the protein, and all-atom docking was performed using the energy minimized structures for all ligands with an effort value of 10. The best-scored docking solutions were further optimized by iterative rounds of minimization and Monte Carlo sampling of the ligand conformation, including the surrounding side-chain residues (within 5 Å of the ligand) and the three water molecules in the KOR orthosteric site. All the above-mentioned molecular modeling operations were performed in ICM-Pro v3.9 molecular modeling and drug discovery suite (Molsoft LLC)[95].

The optimized ligand conformations inside of active state KOR presented on Fig. 1d and Supplementary Fig. 4e–g are provided as supplementary datasets 3 (noribogaine) and 4 (oxa-noribogaine).

**X-ray structure determination.** X-ray diffraction data were collected on a Bruker Apex II diffractometer. The structures were solved by using direct methods and standard difference map techniques and were refined by full-matrix least-squares procedures on F2 with SHELXTL (Version 2014/7)[96–98]. Crystallographic data have been deposited with the Cambridge Crystallographic Data Centre (CCDC #2215015).

The X-ray structure and checkcif report are provided as supplementary datasets 1 and 2.

**Synthesis.** Reagents and solvents (including anhydrous solvents) were obtained from commercial sources and used without further purification unless otherwise stated. All compounds, except noribogaine, were prepared in racemic form. All reactions were performed in oven- or flame-dried glassware under an argon atmosphere unless otherwise stated and monitored by TLC using solvent mixtures appropriate to each reaction. All column chromatography was performed on silica gel (40–63 μm). Preparative TLC was performed on 20 × 20 cm plates with a 1 mm silica layer. For compounds containing basic nitrogen, triethylamine (Et$_3$N) was often used in the mobile phase to provide better resolution. In these cases, TLC plates were pre-soaked in the Et$_3$N containing solvent and then briefly dried before use, to yield an accurate R$_f$ value. Nuclear magnetic resonance spectra were recorded on Bruker 400 or 500 MHz instruments as indicated, using Bruker TopSpin (3.5 pl6) software. Spectra were further analyzed using MestReNova 14. Chemical shifts are reported as δ values in ppm referenced to CDCl$_3$ ($^1$H NMR = 7.26 and $^{13}$C NMR = 77.16). Multiplicity is indicated as follows: s (singlet); d (doublet); t (triplet); q (quartet); p (pentet); dd (doublet of doublets); ddd (doublet of doublet of doublets); dt (doublet of triplets); td (triplet of doublets); m (multiplet); br (broad). For compounds containing a carbamate group, complex spectra with split peaks were observed (presence of conformers). As a result of these effects, multiple peaks may correspond to the same proton group or carbon atom. When possible, this is indicated by an "and" joining two peaks or spectral regions. Alternatively, when certain carbon peaks overlap and thus represent two carbons they were indicated by (2 C) designation. For easier identification of carbamate containing compounds on the included $^1$H NMR spectra, groups of peaks arising from the hindered rotation were integrated together to obtain whole number integrals. All carbon peaks are rounded to one decimal place. Low-resolution mass spectra were recorded on either a JEOL LCmate (ionization mode: APCI+) or Advion expression-L CMS quadrupole mass spectrometer equipped with ESI and APCI sources. High-resolution mass spectra (HRMS) were acquired on a high-resolution Waters XEVO G2-XS QToF mass spectrometer equipped with a UPC2 SFC inlet, on-board fluidics and an ESI probe.

Synthetic conditions and compound characters are provided in the Supplementary information file.

**Reporting summary**
Further information on research design is available in the Nature Portfolio Reporting Summary linked to this article.

## Data availability
The crystallographic data (oxa-ibogaine **10a**) generated in this study have been deposited in the Cambridge Crystallographic Data Centre database under accession code #2215015 www.ccdc.cam.ac.uk/structures and are provided together with the docking structures as supplementary datasets 1-4.

The data generated in this study are provided in the Supplementary Information and Source Data files. Source data are provided with this paper.

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

## Acknowledgements

This work was supported by Columbia University (CU, D.S.), High Point University (HPU, S.H.), the National Institute on Drug Abuse (NIDA) of the National Institute of Health (NIH), grants R01DA050613 (D. Sames and S. Hemby), R33DA045884 (S.M.), R33DA038858 (V. Katritch), and the National Institute of Mental Health (NIMH), grant R21MH116462 (J.P.), and the G. Harold and Leila Y. Mathers Foundation (D.S. and V.H.), and the Hope for Depression Research Foundation (J.A.J.). This research was funded in part through the NIH/NCI Cancer Center Support Grant P30 CA008748 to MSKCC. V.H. acknowledges the Experientia Foundation Postdoctoral Fellowship. The authors would like to thank William Nguyen, Ky Truong, Lana Rasoul, Alexa Stafford, Guy Page and Dr. Richard Kondo (AnaBios Corporation) for technical assistance and helpful discussions, Dr. Fereshteh Zandkarimi (CU) for measuring the high-resolution mass spectra of compounds and to Prof. Gerard F. Parkin and Dr. David A. Sambade (CU) for solving the X-ray crystal structure of oxa-ibogaine. Min H. (Jimmy) Kyaw (CU) reproduced the palladium-catalyzed cyclization procedure under supervision by V.H. A.H. was supported by the Summer Research fellowship (SuRF) at HPU. We thank Dr. Ignacio Carrera (Universidad de la República, Uruguay) for helpful discussions on iboga pharmacology, and the Psychoactive Drug Screening Program (PDSP) at UNC at Chapel Hill for performing receptor panel screening. Certain graphics (on Figs. 2a, 5a) were created using BioRender.com. Research reported in this publication was supported by the Office of The Director, National Institutes of Health of the National Institutes of Health under Award Number S10OD026749. The content is solely the responsibility of the authors and does not necessarily represent the official views of the National Institutes of Health.

## Author contributions

D.S. conceptualized and supervised the work. S.E.H. designed, guided and supervised the CPP tests in mice, neurotrophin protein brain level determination in rats, self-administration and reinstatement studies in rats, OIH studies, statistical analyses and interpretation of the corresponding results. V.H. contributed to the design, performed scale-up synthesis, developed palladium-catalyzed synthetic approach, supervised pharmacological characterization including data interpretation and summarized the collected data for publication. A.C.K. contributed to the design, synthesis, and pharmacological characterization of compounds, including the design and first synthesis of oxa-noribogaine, carried out early synthetic work including reaction development, optimization, and compound characterization and supervised initial pharmacological characterization and data interpretation. B.B. contributed to the scale-up synthesis, contributed to the design and performed mice experiments conducted at Columbia University (tail-flick, open field, and forced swim tests) and interpreted the corresponding results. S. McIntosh and L.S. carried out self-administration and conditioned place preference experiments performed at HPU. A.H. trained rats for self-administration studies and conducted the food maintained responding experiments. M.G.W and M.N. performed the BRET functional assays. A.H. performed the opioid binding assays. M.A. performed the tail-flick mice assay comparing WT and KO mice and J.E.P. supervised the tail-flick mice assays with KO mice. C.H. carried out the serotonin transporter inhibition assay. R.S.O. performed the TRUPATH assays. N.A.-G. provided expert consultation for cardiotoxicity studies. S.A.Z. carried out the docking studies and V.K. supervised the docking studies. M.Y. oversaw the in vivo experiments at Columbia University and provided expert guidance. J.A.J. supervised the BRET functional assays and interpretation of the corresponding results. S. Majumdar supervised the binding assays, functional assays and early in vivo mice tests (at MSKCC) and TRUPATH assays (at WUSTL). D.S. wrote the manuscript with significant help from V.H. and S.E.H. All authors contributed to the editing of the manuscript.

## Competing interests

V.H., A.C.K, B.B., M.G.W, J.A.J., S.E.H and D.S. are named inventors on a patent(s) related to oxa-iboga compounds. A.C.K and D.S. are co-founders of Gilgamesh Pharmaceuticals, which licensed the oxa-iboga assets from Columbia University. All other authors declare no competing interests.

## Additional information

[1]Department of Chemistry, Columbia University, New York, NY 10027, USA. [2]Department of Basic Pharmaceutical Sciences, Fred Wilson School of Pharmacy, High Point University, High Point, NC 27268, USA. [3]Department of Molecular Pharmacology and Therapeutics, Columbia University, New York, NY 10032, USA. [4]Department of Psychiatry, Columbia University, New York, NY 10032, USA. [5]Division of Molecular Therapeutics, New York State Psychiatric Institute, New York, NY 10032, USA. [6]Department of Neurology and Molecular Pharmacology, Memorial Sloan Kettering Cancer Center, New York, NY 10065, USA. [7]Center for Clinical Pharmacology, University of Health Sciences & Pharmacy at St Louis and Washington University School of Medicine, St Louis, MO 63110, USA. [8]Department of Neuroscience and Cell Biology, Rutgers University, Piscataway, NJ 08854, USA. [9]Rutgers Addiction Research Center, Brain Health Institute, Rutgers University, Piscataway, NJ 08854, USA. [10]Washington University Pain Center and Department of Anesthesiology, Washington University School of Medicine, St Louis, MO 63110, USA. [11]AnaBios Corporation, 1155 Island Ave, Suite 200, San Diego, CA 92101, USA. [12]Department of Quantitative and Computational Biology, University of Southern California, Los Angeles, CA 90089, USA. [13]Department of Chemistry, Bridge Institute, Michelson Center for Convergent Sciences, University of Southern California, Los Angeles, CA 90089, USA. [14]Mouse Neurobehavioral Core facility, Columbia University Irving Medical Center, New York, NY 10032, USA. [15]The Zuckerman Mind Brain Behavior Institute at Columbia University, New York, NY, USA. [16]These authors contributed equally: Václav Havel, Andrew C. Kruegel, Benjamin Bechand. ✉e-mail: ds584@columbia.edu

