## [Peer Review File · Nature Communications]

REVIEWER COMMENTS

Reviewer #1 (Remarks to the Author):

The manuscript by Havel et al., report the development of novel modified iboga alkaloid molecules that have improved efficacy to attenuate opioid seeking in rodent self-administration models and lack the proarrhythmic effects of ibogaine and noribogaine.

The complex structure of ibogaine and noribogaine generates complex pharmacology via targeting of NMDA-R, nicotinic-R, SERT, KOR, but not 5HT_{2A}-R. Ibogaine alkaloids have intriguing properties suggesting potential beneficial use as therapeutic agents for drug addiction, but due to serious cardiac adverse effects including sudden death, such use has not materialized.

The authors have developed new synthetic methods to generate the iboga scaffold that enabled preparation of the analogs, oxa-noribogaine and epi-oxa-noribogaine.

1. It is unclear how the authors zero in on KOR agonism for the oxa-iboga compounds, since if simply based on structural comparisons with the iboga compounds, this would not be intuitive.
2. Results show oxa-noribogaine had markedly increased KOR activity compared to noribogaine – with higher KOR binding affinity, G protein activation efficacy via 35SGTP γ S and BRET assays, all in nanomolar ranges, even though the essential components of the KOR agonist pharmacophore is identical in both the iboga and oxa-iboga molecules. How do the authors explain the marked differences in KOR activity between the two compound classes? In iboga and oxa-iboga pharmacophores, binding to KOR seem to involve the same residues in both classes. This would not explain their differential affinities – it would be helpful if mutagenesis of the proposed sites of interaction could be used to show potential differences.
3. Noribogaine has been shown to have MOR antagonist activity, in addition to KOR agonist activity. Therefore, is it possible that some of the in vivo effects of oxa-noribogaine could be due to actions at MOR, as well as on opioid receptor heterodimers MOR/KOR, MOR/DOR, or KOR with other GPCRs? These possibilities should be evaluated.
4. The receptor mechanism by which oxa-noribogaine induces potent analgesia should be elucidated, since it was equipotent with the full KOR agonist U50,488. The lack of complete attenuation of oxa-noribogaine analgesia in KOR-ko mice (<1 log unit shift in dose response curve) and the shift in its analgesic potency in MOR-ko mice make it imperative to consider oxa-

noribogaine effects on heterodimeric complexes involving KOR, as ligand binding to one GPCR protomer can allosterically regulate affinity at its dimer partner. Furthermore, what GPCR mechanism could account for only partial agonist activity at the level of receptor, G proteins and beta-arrestin but signal amplification that results in full agonist activity at downstream effectors to generate full functional effects such as potent analgesia and robust elevation of GDNF?

5. Since the oxa-iboga compounds were primarily characterized as partial KOR agonists, the analysis should be extended to evaluate the many signaling pathways involved in KOR action they may activate in addition to G proteins and beta-arrestin2, such as GRK3, GIRK, p38 MAPK, mTOR etc. The phosphorylation state of specific KOR residues in the C-terminus and effects on receptor internalization after activation by oxa-noribogaine compared to U50,488 should be investigated to provide insight into mechanisms of action resulting in the functional effects reported.

6. In Discussion, use of "intermediate partial signaling efficacy" to describe oxa-noribogaine falls short of the paradoxical signaling and functional discrepancies of the results presented.

7. In summary, the generation of the oxa-iboga compounds is noteworthy, and the results presented on their novel properties are intriguing. However, the complete absence of any in vitro or in vivo investigations that shed light on the pharmacological mechanisms underlying these effects seriously detract from the impact and result in a descriptive exercise.

Reviewer #2 (Remarks to the Author):

This is an interesting report of an ibogaine analogue with reduced cardiovascular effects and the ability to suppress morphine, heroin, and fentanyl self-administration. The work is noteworthy and significant to the field. The methodology appears scientifically sound the results are likely to generate interest in the field. As such, I feel it is worthy of publication after the following issues are addressed:

1. The term "kappa psychedelic" is curious at best. The term psychedelic is used for agents which interact with serotonin 5-HT_{2A} receptors. As the authors show, oxa-noribogaine does not. Multiple reports also show that the effects of a serotonergic hallucinogen are very different than a kappa opioid agonist. This is also a curious choice when the authors cite a human study of kappa agonists producing "psychotomimetic" effects (ref 31). In the discussion, the authors cite ref 21 but this is likely a error. If the authors have additional citations that clearly describe the behavioral differences of a "kappa psychedelic" from a classical psycedelic, these need to be provided.

2. The authors results with adult primary human hearts are viewed as a strength of the work. However, the discussion the potential mechanism is unfulfilling. While the reviewer agrees that more detailed follow-up studies are outside the scope of the present work, a greater discussion of the ion channels profiled in the broad panel screen should be included. More enthusiasm would be garnered if a set of common ion channels were profiled as well. For example, the hERG channel.

Reviewer #3 (Remarks to the Author):

The authors describe a novel class of modified ibogaine fragments defined by the replacement of indole with benzofuran (oxygen ring substitution). Interestingly, some natural and synthetic approved drugs contain a benzofuran unit, such as Amiodarone (Antiarrhythmic drugs), Ramelteon (Sedative drug), Vilazodone and Citalopram (Anti-depressive drugs).

The study compares oxa-ibogaine to its demethylated congener or oxa-noribogaine and describes their synthesis and mechanism of action in vitro and nonclinical animal studies. The authors report that oxa-noribogaine is superior to its ibogaine congener. Both drugs lack cardiac risks and lead to lasting blockade of opioid taking behaviors in animal models. In addition, the drugs appear to lack abuse liability in keeping with what has been reported for ibogaine and noribogaine in animal models.

This manuscript describes a large series of well-designed experiments to define the opportunities of advancing oxa-iboga compounds. There are some specific comments and queries provided to the authors for review and comment listed below. The authors are encouraged to provide some additional comment in the discussion of limitations on whether they plan to further optimize the full potential of benzofuran iboga compounds for selectivity and multifunctional on- and off-target opportunities.

1.) The authors state that noribogaine was synthesized as previously described starting from Voacangine isolated from root bark of Voacanga Africana (their prior study; see ref 35.). Also, voacangine was isolated from the tree bark using the method of Jenks, C. W. (2002) Extraction studies of Tabernanthe iboga and Voacanga africana. Nat. Prod. Lett. 16, 71–76. Since this first study is expected to draw wide attention, the reviewer suggests a few additional details should be included since noribogaine is a lead drug for this series.

2.) The conversion of voacangine to noribogaine has been reported as early as 1957 (Janot and Goutarel, US 2,813,873). However, it is unclear if this was done in either a one-step process in

going from voacangine to noribogaine (3) using HOAc/HBr (48%, reflux) without separation of any intermediates, or via a two-step process starting with converting voacangine (1) to Ibogaine (KOMe), followed by converting the ibogaine to Noribogaine (3) (HBr, 48% /HOAc/ reflux). Specifically, the authors should state that this does not involve the intermediacy of ibogaine. The lithium salt of voacangine can be prepared by treating voacangine with n-butyllithium in hexane at 0 °C with 1-propanethiol (see, Kuehne, et al. J. Med. Chem., 2003, 46, 2716-2730). If these prior studies are relevant, they should perhaps be cited. The authors should state that desmethyl-voacangine is the starting material.

3.) The dose formulation for the animal studies and aqueous solubility and stability of the compounds tested should be further stated in the methods if known. Lipophilicity parameters?

4.) The results emphasize the importance of partial kappa (KOR) agonism as a prototype mechanism mediating decreased opioid self-administration, analgesia and other effects in animal models. The receptor pharmacology of ibogaine vs. noribogaine was recently summarized for binding potency between KOR and non-opioid human targets including SERT and $\alpha 3\beta 4$ molecular targets in recent review article (DOI: 10.1016/j.phrs.2022.106620). This citation provides references to the earlier studies which are omitted in the current report and should be included for the reader.

5.) In keeping with prior reports by Dorit Ron and more recently, Olsen and coworkers on noribogaine's effect on GDNF expression, administration of oxa-noribogaine (40 mg/kg; i.p.) significantly increased GDNF protein levels compared to control group. However, the magnitude compared to noribogaine was massively larger (100 – 200% increase). The authors suggest that targeted elevation of GDNF in dopamine regions of the brain underscores the improved efficacy compared to the lead molecule. Aticaprant blocks the elevation consistent with a kappa mediated effect. This part of the study is very important for defining a novel MoA for iboga alkaloids and effects of ibogaine in humans undergoing detoxification from opioids and other drugs. These studies alone warrant expedited publication of their work.

6.) One potential concern is potential for GDNF mediated neurotoxic effects in the cerebellum. The authors do not state if the cerebellum was examined in their study (perhaps the authors have kept the brains). Comment.

7.) GDNF has been linked to cerebellar toxicity (Luz et al., 2016) in keeping with the original work of Mark Moliver and others who demonstrated toxicity to cerebellar Purkinje neurons following high doses of ibogaine. The Purkinje cells express GDNF. This observation may warrant some discussion by the authors since as they know, the GDNF milestone started as a hypothesis for Parkinson's disease, too.

8.) Since nigrostriatal GDNF overexpression induces a robust weight loss in both animal models and clinical trials, the authors should comment on whether there was an observed food intake given the long-lasting effects of single dose administration in their animal models. Targeted expression up to two-fold increases may lead to ventricular spillover. A potential off target effect paragraph should be added in the discussion which includes GDNF.

9.) The lack of pro-depressive or aversive behavioral effects are not induced by oxa-noribogaine is an important observation and strongly supportive of the authors' conclusions and enthusiasm

overall for this class of new compounds. However, this reviewer was unclear what accounts for this improved tolerability (e.g., GDNF – dopamine vs. serotonin reuptake). This is not overly clear. Multi-target vs. single KOR-GDNF mechanism of action.

10.) The long lasting pharmacodynamic effect of oxa-noribogaine makes this reviewer wonder if the drug demonstrates pseudoirreversible binding to KOR. Was this tested or under review?

11.) This reviewer is intrigued by the pharmacokinetics of the compounds and the methods need more details on the sampling times and rationale for selection of s.c. and i.p. doses. Is there more known from dose range finding studies that are not included in this report?

12.) What is the inferred metabolic pathway for this class? Is there an active metabolite which accounts for the rapid clearance in plasma and long aftereffects? Has this been examined in vitro?

13.) Is oxa-ibogaine converted to oxa-noribogaine in vivo? Or does the oxygen substitution decrease CYP2D6 metabolism? Figure 3 shows rapid clearance from plasma and brain. Wash out appears to be complete in 5 hours. All of the lasting after effects are GDNF mediated? The pharmacokinetic parameters should be better described in the methods and results linked to the time course of behavioral effects. This reviewer found it difficult to decipher from a read of the methods section.

14.) A primary observation of the current report is that oxa-iboga compounds do not show any evidence of pro-arrhythmia risks in adult primary human heart cells. Although there is much discussion of hERG, there was discussion of actual binding to or hERG block measurements? PSP screen negative at hERG?

15.) The discussion of compensatory effects induced by the enhanced inhibition of L-type calcium channels in this species versus human (Koenig et al., reference) is outdated. In vivo QTc prolongation mitigation by calcium channel block requires very similar affinities for hERG compared to calcium (e.g., verapamil). Ibogaine does not significantly occupy calcium channels making this explanation unlikely. However, there are no data (unless it was missed by this reviewer) which informs calcium vs. hERG channel specific effects of ibogaine or oxa-ibogaine. Comment or clarify further in the results.

16.) Of note, the Glue et al., (2016) paper was done in opioid dependent patients administered noribogaine 2 hours after morphine to patients who likely had residual methadone in blood (see limitations sections of the manuscript). Thus, the slope of the fit for estimating QTc vs. ng/ml noribogaine may not be conclusive in this study in contrast to what is stated by the authors.

17.) Shi et al., (2021; ALTEX 38(4), 636-652. doi:10.14573/altex.2103311) provides an updated review of the modeling to predict cardiotoxicity. See – figure 8 comparing ibogaine to noribogaine. Therapeutic free equivalents are somewhat difficult to estimate based on a number of factors. Figure 4 shows the data for noribogaine compared to oxa-noribogaine. Was ibogaine tested in this assay? If so, the results should be included because the lead up discussion to figure 4 describes the risk associated with ibogaine. The narrative is leading the reader to conclude that ibogaine is safer than noribogaine; however metabolic ratios and in vivo electrocardiology are an emerging and still understudied area. Thus, conclusions at this juncture should be limited to the context of the authors study design, results and their relevance for a new class of modified iboga compounds.

18.) The authors state that “most of the reported adverse effects have occurred > 24 hours post ibogaine ingestion (ref. 34), noribogaine with its long circulation and large exposure appears to be the culprit of cardiac risks.” Ibogaine has saturable first pass metabolism following a single dose. A review suggests that there is a range from the reported time between ibogaine ingestion and death of approximately 24 hours (range 1.5–76 hours; Luz and Mash, 2021). High doses of ibogaine, stacked and impure drug substance along with other factors which confer risk (K⁺ and Mg⁺⁺ deficits) are unknowns.

19.) The time course for most of these cases are uncertain and blood levels and toxicology were not available. See also ref. 14 the report from Aćimović et al. illustrates the difficulties when assessing the causality of ibogaine fatalities based on incomplete information from forensic and toxicologic investigations. Very high ibogaine concentrations were measured in femoral blood (3,260 ng/mL) and urine (28,870 ng/mL) and died 5–12 hours after oral ingestion. Thus, a definite causality statement for assigning toxicity to the metabolite or parent drug seems to this reviewer to be outside the scope of the current study. The lack of any effect in the model assay for oxa-iboga alkaloids is relevant for future drug design to limit off target effects.

Reviewer #4 (Remarks to the Author):

Ibogaine has had an interesting and important history for potential treatment of SUD but use has been hampered by cardiac toxicity. It has been recognized that Ibogaine has many psychotropic targets that include kappa opioid receptors as well as many other receptors and channels with diverse psychoactive inferences. Using innovative synthetic approaches, the authors present a large body of work characterizing the pharmacology of the novel ibogaine derivatives, oxa-noribogaine and epi-oxa. Oxa-noribogaine showed greater efficacy at the kappa opioid receptor (KOR) than noribogaine and greater efficacy than noribogaine in suppressing morphine self-administration. Importantly, unlike noribogaine (10 µM), oxa-noribogaine (10 µM) did not show pro-arrhythmia risks in human adult primary heart cells. Oxa-noribogaine (10 mg/kg) in rats reduced self-administration and reinstatement to both morphine and fentanyl, and unlike cocaine (10 mg/kg) and morphine (20 mg/kg), did not induce conditioned place preference at 10 and 40 mg/kg doses in mice. In addition, oxa-noribogaine, showed comparable analgesic effects to a full KOR agonist in mice, without reducing locomotor activity in the open field test. The novel compound increased glial-derived neurotrophic factor (GDNF) protein in the mPFC and VTA, but not the NAc.

Concerns:

1) Ibogaine is a promiscuous drug with regard psychoactive targets and though kappa agonists are hallucinogenic and oppose reward in many assays, care in presumption that the kappa activity is mediating the effects relevant to SUD should be taken.

2) The findings in the manuscript are limited by the fact that there are discrepancies between the doses and drug comparisons across the various experiments and between mice and rats. This

prevents generalizability of the findings across the experiments. For instance, the finding that oxa-noribogaine (40 mg/kg) suppresses morphine intake in rats and 10mg/kg suppresses relapse is uncoupled from the finding that oxa-noribogaine (5.4 mg/kg) in mice does not suppress locomotion. Hypolocomotion/sedation is a major problem with interpreting behavioral assays following kappa agonist treatments and could explain a portion of the data. This discrepancy needs to be carefully addressed throughout the manuscript and perhaps an oxa-noribogaine dose response for sedation and analgesia in both mice and rats would help alleviate this potential confound.

3) Though there was a substantive kappa-dependent increase in GDNF protein levels in the VTA from oxa-noribogaine, what this means for which of the behavioral sequelae was unclear and not tested. The consequence of up or down regulation of GDNF in the VTA seems drug-specific and there are many questions in this reviewer's mind as to how this finding provides mechanistic substance to the paper. Some additional context re interpretation of the regional specificity in regulation of GDNF would help the reader.

4) The oxa-noribogaine analgesia, unlike the Epi-Oxa appears to have a mu and kappa component based on the KO as well as the antagonist data in supplementary. This makes pharmacology complex. Does oxa-noribogaine have sedation at an EC80 for analgesia in the Mu KO mouse? Also, for completion it would be important to see the effects on oxa-noribogaine analgesia in the females in the Mu KO mouse (Fig 3e) since in most mouse strains females are considerably less sensitive to kappa agonist analgesia than males. Lastly, that kappa-agonist induced analgesia could have a stress-induced component should be discussed– this would not be a “desirable therapeutic-like effect” as indicated in the manuscript.

5) Unconvincing is that oxa-noribogaine is working as an atypical kappa agonist as opposed to a kappa/mu agonist/PAM. This possibility is based upon the overlapping kappa properties of oxa-noribogaine and epi-oxa in the invitro kappa assays yet very different in vivo pharmacological properties (sedation and analgesia). Also the analgesia data show a mu component for oxa-noribogaine but not epi-oxa in the KO data and in fig S6, which shows the mu antagonist more effective than kappa antagonist at blocking oxa-noribogaine analgesia. Finally there looks like a strong tendency for oxa-noribogaine to be rewarding in the CPP assay at 40mg/kg that statistically may not be different from the morphine results. Pure kappa agonists would be aversive in the CPP assay and I would assume epi-oxa is aversive though this data was not found in the manuscript.

6) It would be important to know if oxa-noribogaine has any affinity for the ORL-1 receptor since ORL-1 agonists can modify rewarding actions. Oxa-noribogaine was tested in the primary screen shown in a supplementary figure but hazy as to if there was significant activity/binding.

7) The discussion dismisses sex differences in oxa-noribogaine behaviors which should be cautioned – many kappa-regulated behaviors are markedly different between sexes though not so dramatically with mu.

Minor:

1. Consider removing the vague, “radically different,” from the first sentence of the abstract, so that it simply reads “...prototypes of treatment of substance use disorders...”

3. The final sentence of the first paragraph on line 100, beginning with “To further probe” was unspecific and difficult to interpret until the next paragraph was read. To improve, consider more specific language for this sentence, such as “To determine whether the KOR signaling initiated by oxa-noribogaine is comparable to a full KOR agonist, we examined... which indicated...”
4. The section header on line 190 omits sedative effects. This adds to some confusion once the paragraph on sedation comes up, especially as the leading sentence leads with “hallucinosis” which is misleading to the reader as to the anticipated content within the paragraph.
5. In all figures the drug doses in mg/kg for all drug conditions should be included for each figure panel and in the figure legend – the species used should be clear.
6. The inclusion of imipramine in Fig. 3i could be further explained. The drug is mentioned just once as a monoamine reuptake inhibitor in the Introduction. Is this drug and dose intended to result in the same “pro-depressive-like effects” found after standard kappa psychedelics as described in Discussion?

REVIEWER COMMENTS

Reviewer #1 (Remarks to the Author):

The manuscript by Havel et al., report the development of novel modified iboga alkaloid molecules that have improved efficacy to attenuate opioid seeking in rodent self-administration models and lack the proarrhythmic effects of ibogaine and noribogaine.

The complex structure of ibogaine and noribogaine generates complex pharmacology via targeting of NMDA-R, nicotinic-R, SERT, KOR, but not 5HT2A-R. Ibogaine alkaloids have intriguing properties suggesting potential beneficial use as therapeutic agents for drug addiction, but due to serious cardiac adverse effects including sudden death, such use has not materialized. The authors have developed new synthetic methods to generate the iboga scaffold that enabled preparation of the analogs, oxa-noribogaine and epi-oxa-noribogaine.

1. It is unclear how the authors zero in on KOR agonism for the oxa-iboga compounds, since if simply based on structural comparisons with the iboga compounds, this would not be intuitive.

This was a discovery via a systematic SAR work. Over the years of examining iboga pharmacology, we have observed consistently that a benzofuran substitution of the indole leads to enhanced KOR binding and signaling.

2. Results show oxa-noribogaine had markedly increased KOR activity compared to noribogaine – with higher KOR binding affinity, G protein activation efficacy via 35SGTPgammaS and BRET assays, all in nanomolar ranges, even though the essential components of the KOR agonist pharmacophore is identical in both the iboga and oxa-iboga molecules. How do the authors explain the marked differences in KOR activity between the two compound classes? In iboga and oxa-iboga pharmacophores, binding to KOR seem to involve the same residues in both classes. This would not explain their differential affinities – it would be helpful if mutagenesis of the proposed sites of interaction could be used to show potential differences.

We agree with this reviewer. The rationale for the difference in binding potency is not clear from the docking studies. We did not embark on the mutagenesis studies, despite the expertise on our team of co-authors, as the binding poses are essentially the same between the benzofuran and the indole iboga compounds (supplementary Fig. S4e-g). We expect the observed differences in binding potency are due to electronic effects, namely changes in the electron density distribution in the ligand ring and resulting interactions with larger surfaces of the receptor. These kinds of electronic effects are often hard to rationalize by mutagenesis studies and are beyond the scope of this study. We have added a brief discussion remark to this effect in the revised manuscript in the main text and Supplemental Information.

3. *Noribogaine has been shown to have MOR antagonist activity, in addition to KOR agonist activity. Therefore, is it possible that some of the in vivo effects of oxa-noribogaine could be due to actions at MOR, as well as on opioid receptor heterodimers MOR/KOR, MOR/DOR, or KOR with other GPCRs? These possibilities should be evaluated.*

The role of MOR has been characterized both *in vitro* and *in vivo*; for the latter via both genetic and pharmacological tools. In response to this reviewer's comments, we now also add new data on MOR signaling efficacy via two different functional readouts in two species (mouse and human). We provide a reasonable hypothesis for the observed *in vivo* pharmacology, please see below for more details.

4. *The receptor mechanism by which oxa-noribogaine induces potent analgesia should be elucidated, since it was equipotent with the full KOR agonist U50,488. The lack of complete attenuation of oxa-noribogaine analgesia in KOR-ko mice (<1 log unit shift in dose response curve) and the shift in its analgesic potency in MOR-ko mice make it imperative to consider oxa-noribogaine effects on heterodimeric complexes involving KOR, as ligand binding to one GPCR protomer can allosterically regulate affinity at its dimer partner. Furthermore, what GPCR mechanism could account for only partial agonist activity at the level of receptor, G proteins and beta-arrestin but signal amplification that results in full agonist activity at downstream effectors to generate full functional effects such as potent analgesia and robust elevation of GDNF?*

This reviewer raises an interesting point about the potential amplification mechanisms for partial agonists. One potential mechanism involves downstream amplification via catalytic effectors such as kinases (Boltaev, U. et al. *Science Signaling* 2017, DOI: 10.1126/scisignal.aal1670). We have now added a comment to this effect in the revised manuscript (Discussion and Limitations). We respectfully disagree with this reviewer about the need to examine the heterodimeric complexes. Despite the expertise of our team of collaborators (Dr. Jonathan Javitch), this direction is beyond the scope of this work, which introduces a novel and promising agent for treatment of substance use disorders and already contains large amounts of data, including the relative contribution of KOR and MOR to the antinociceptive effects supported via both genetic and pharmacological methods.

5. *Since the oxa-iboga compounds were primarily characterized as partial KOR agonists, the analysis should be extended to evaluate the many signaling pathways involved in KOR action they may activate in addition to G proteins and beta-arrestin2, such as GRK3, GIRK, p38 MAPK, mTOR etc. The phosphorylation state of specific KOR residues in the C-terminus and effects on receptor internalization after activation by oxa-noribogaine compared to U50,488 should be investigated to provide insight into mechanisms of action resulting in the functional effects reported.*

This request also represents a separate study on its own and is beyond the scope of an already extensive paper. Further, the western blot (WB) technique lacks the robust dynamic range in terms of efficacy or extent of phosphorylation (see for example Boltaev,

U. et al. Science Signaling 2017, DOI: 10.1126/scisignal.aal1670). We are aware that not everyone agrees on this point as WBs are heavily used due to the lack of more robust methods. Nevertheless, we have added this point to the Limitations section, as the detailed phospho-proteomic analysis is warranted in the future studies.

6. In Discussion, use of “intermediate partial signaling efficacy” to describe oxa-noribogaine falls short of the paradoxical signaling and functional discrepancies of the results presented.

We agree overall. However, partial agonism is a reasonable hypothesis, now strengthened by the new MOR signaling data (supplementary Fig. S5). Namely, MOR does not contribute to the antinociceptive effects of epi-oxa in mice *in vivo*, in contrast to oxa-noribogaine where it plays a minor contribution. Both compounds have comparable binding and signaling potency at MOR *in vitro*, but epi-oxa is markedly less efficacious in the MOR BRET assays and inactive (below detection limit) in the MOR Nb33 sensor assays (in both mouse and human species examined). Thus, we suggest that there is a signaling threshold for the upstream signaling, to translate into an *in vivo* pharmacodynamic effect like antinociception. In other words, the signaling efficacy hypothesis explains the *in vivo* pharmacological differences between oxa-noribogaine and epi-oxa. The latter compound is not sufficiently efficacious at MOR to contribute to the *in vivo* pharmacological profile. We also proposed that the partial KOR agonism is a reasonable hypothesis, not a definitive model, for the observed differences between the oxa-iboga compounds and standard KOR agonists, while we provide alternative hypotheses.

In the broader context, partial signaling efficacy hypotheses have been proposed for other classes of psychoactive compounds with unusual *in vivo* effects (e.g. non-hallucinogenic 5HT_{2A} agonists, Cunningham, M.J. et al., 2022, *ACS Chem. Neurosci.*, DOI: 10.1021/acscchemneuro.2c00597; Lewis et al., 2023, *Cell Reports*, DOI: 10.1016/j.celrep.2023.112203)

We have added a new discussion to this effect, to temper and qualify our proposed model, while acknowledging the limitations and potential future studies suggested by this reviewer. We specifically added the idea of KOR-GPCR heterodimers as an alternative explanatory hypothesis for future studies in both the Discussion and Limitation sections.

7. In summary, the generation of the oxa-iboga compounds is noteworthy, and the results presented on their novel properties are intriguing. However, the complete absence of any in vitro or in vivo investigations that shed light on the pharmacological mechanisms underlying these effects seriously detract from the impact and result in a descriptive exercise.

We are glad that this reviewer found the work interesting. Unfortunately, we find the following statement unreasonable.

“However, the complete absence of any in vitro or in vivo investigations that shed light on the pharmacological mechanisms underlying these effects seriously detract from the impact and result in a descriptive exercise.”

We would kindly ask this reviewer to see the amount of data in the main text, experimental and supplemental sections. The manuscript describes extensive *in vitro* pharmacology characterization (panels of GPCR and other targets, classic and state-of-the-art binding and functional assays, the entire repertoire of Galpha protein isoforms, beta-arrestin signaling, receptors of two species), *in vivo* pharmacology (antinociception in KOR/MOR KOs, pharmacological inhibition studies), *in vivo* behavior (CPP/CPA, FST, locomotion), as well as *in vivo* down-stream signaling effects (GDNF in different brain regions). It is not clear to us what “the complete absence of any *in vitro* and *in vivo* investigations” really means.

Taking a larger view, the paper covers a wide range of new chemistry, GPCR *in vitro* and *in vivo* pharmacology and behavior, and rodent SUD-relevant behavior. We are introducing a novel class of compounds with intriguing pharmacology, improved cardio safety, and promising therapeutic potential. Yes, we would love to know the answers to all the interesting questions this reviewer raised, but each goes deep into a separate study.

At the same time, we understand and appreciate the thrust of this reviewer’s comments. We therefore tempered the relevant claims, starting in the title and the abstract, by removing “with defined molecular signaling”. We revised the title of the entire study to capture the main story – introduction of a novel, promising agent, with solid preliminary pharmacological characterization. In the revised manuscript, we also provide a reasonable explanatory model, supported by the experimental data, while amply qualifying its limitations.

Lastly, we would like to ask this reviewer to consider the fact that iboga pharmacology is very complex. Therefore, the usual “linear” mechanistic considerations that apply to relatively selective compounds are not necessarily productive. In the case of oxa-iboga, we “pulled out” KOR/MOR from the iboga pharmacological bundle, which allowed us to determine which molecular targets drive the antinociceptive effects as a PD measure of KOR/MOR engagement *in vivo*. Up to this point, we provide solid characterization of KOR/MOR’s relative contributions. Not on the level of potential heterodimers, but with clear relative contribution of each receptor *in vivo*. Going further to fully explain the range of atypical features of the oxa-iboga compounds would necessitate long studies with unclear outcomes – well beyond the scope of this already extensive study. For those future studies we provide a leading, reasonable hypothesis based on signaling efficacy (see a similar hypothesis for kratom alkaloids/MOR efficacy: Bhowmik, S. *et al.*, 2021, *Nat Commun*, Doi: 10.1038/s41467-021-23736-2, and psychedelics/5HT2A efficacy: Cunningham, M.J. *et al.*, 2022, *ACS Chem. Neurosci.*, DOI: 10.1021/acscchemneuro.2c00597). We also provide the key caveats for this hypothesis

and potential alternatives suggested by this reviewer. We have revised the Discussion and the Limitation sections as such, to include and acknowledge this reviewer's concerns.

On the experimental front, we have conducted additional MOR signaling experiments that provide a rationale for the different contribution of MOR to the antinociceptive effects of oxa-noribogaine and epi-oxa-noribogaine.

Reviewer #2 (Remarks to the Author):

This is an interesting report of an ibogaine analogue with reduced cardiovascular effects and the ability to suppress morphine, heroin, and fentanyl self-administration. The work is noteworthy and significant to the field. The methodology appears scientifically sound the results are likely to generate interest in the field. As such, I feel it is worthy of publication after the following issues are addressed:

We thank this reviewer for the positive feedback.

1. *The term "kappa psychedelic" is curious at best. The term psychedelic is used for agents which interact with serotonin 5-HT_{2A} receptors. As the authors show, oxa-noribogaine does not. Multiple reports also show that the effects of a serotonergic hallucinogen are very different than a kappa opioid agonist. This is also a curious choice when the authors cite a human study of kappa agonists producing "psychotomimetic" effects (ref 31). In the discussion, the authors cite ref 21 but this is likely a error. If the authors have additional citations that clearly describe the behavioral differences of a "kappa psychedelic" from a classical psychedelic, these need to be provided.*

There are different views on the definition of the term "psychedelic". The term was introduced to describe broadly mind-expanding and mind-altering effects. We prefer this broader definition (this nomenclature was also adopted by others; e.g., Calvey, T. & Howells, F. M., 2018, Progress in Brain Research, DOI: 10.1016/bs.pbr.2018.09.013). Nevertheless, we removed the term "kappa psychedelic" in the revised manuscript as this paper is not an appropriate platform for such debates.

2. *The authors results with adult primary human hearts are viewed as a strength of the work. However, the discussion the potenial mechanism is unfulfilling. While the reviewer agrees that more detailed follow-up studies are outside the scope of the present work, a greater discussion of the ion channels profiled in the broad panel screen should be included. More enthusiasm would be garnered if a set of common ion channels were profiled as well. For example, the hERG channel.*

In response to this comment, we now include data for hERG using radioligand displacement assay. We found similar binding affinity of oxa-noribogaine to that reported for noribogaine and other iboga alkaloids. A comprehensive screening of the cardiac ion channels is beyond the scope and represents a separate study. Nevertheless, we now

include a discussion in the revised manuscript that the lack of pro-arrhythmia effects are likely due to complex multi-channel interactions, a subject of future studies.

Reviewer #3 (Remarks to the Author):

The authors describe a novel class of modified ibogaine fragments defined by the replacement of indole with benzofuran (oxygen ring substitution). Interestingly, some natural and synthetic approved drugs contain a benzofuran unit, such as Amiodarone (Antiarrhythmic drugs), Ramelteon (Sedative drug), Vilazodone and Citalopram (Anti-depressive drugs).

The study compares oxa-ibogaine to its demethylated congener or oxa-noribogaine and describes their synthesis and mechanism of action in vitro and nonclinical animal studies. The authors report that oxa-noribogaine is superior to its ibogaine congener. Both drugs lack cardiac risks and lead to lasting blockade of opioid taking behaviors in animal models. In addition, the drugs appear to lack abuse liability in keeping with what has been reported for ibogaine and noribogaine in animal models.

This manuscript describes a large series of well-designed experiments to define the opportunities of advancing oxa-iboga compounds. There are some specific comments and queries provided to the authors for review and comment listed below. The authors are encouraged to provide some additional comment in the discussion of limitations on whether they plan to further optimize the full potential of benzofuran iboga compounds for selectivity and multifunctional on- and off-target opportunities.

We thank this reviewer for the positive feedback and useful comments below.

1. *The authors state that noribogaine was synthesized as previously described starting from Voacangine isolated from root bark of Voacanga Africana (their prior study; see ref 35.). Also, voacangine was isolated from the tree bark using the method of Jenks, C. W. (2002) Extraction studies of Tabernanthe iboga and Voacanga africana. Nat. Prod. Lett. 16, 71–76. Since this first study is expected to draw wide attention, the reviewer suggests a few additional details should be included since noribogaine is a lead drug for this series.*

Reference for the procedure used for extraction of voacangine was included in the Supplementary Information (González, B. et al., 2021, ACS Omega, DOI: 10.1021/acsomega.1c00745).

2. *The conversion of voacangine to noribogaine has been reported as early as 1957 (Janot and Goutarel, US 2,813,873). However, it is unclear if this was done in either a one-step process in going from voacangine to noribogaine (3) using HOAc/HBr (48%, reflux) without separation of any intermediates, or via a two-step process starting with converting voacangine (1) to Ibogaine (KOME), followed by converting the ibogaine to Noribogaine (3) (HBr, 48% /HOAc/ reflux). Specifically, the authors should state that this does not involve the intermediacy of ibogaine. The lithium salt of voacangine can be prepared by treating voacangine with n-butyllithium in hexane at 0 °C with 1-propanethiol*

(see, Kuehne, et al. *J. Med. Chem.*, 2003, 46, 2716-2730). If these prior studies are relevant, they should perhaps be cited. The authors should state that desmethyl-voacangine is the starting material.

Noribogaine synthesis scheme was added to the Supplementary Fig. S2. Additional statement indicating that noribogaine was prepared from Voacangine through conversion to 10-hydroxy-coronaridine and subsequent decarboxylation to intentionally avoid handling of the schedule I substance ibogaine was added.

3. *The dose formulation for the animal studies and aqueous solubility and stability of the compounds tested should be further stated in the methods if known. Lipophilicity parameters?*

Extensive study of aqueous solubility and stability was not conducted, these are typically done in the drug development process. Lipophilicity parameters were not experimentally determined, however we provide protein plasma and brain tissue binding data.

4. *The results emphasize the importance of partial kappa (KOR) agonism as a prototype mechanism mediating decreased opioid self-administration, analgesia and other effects in animal models. The receptor pharmacology of ibogaine vs. noribogaine was recently summarized for binding potency between KOR and non-opioid human targets including SERT and $\alpha3\beta4$ molecular targets in recent review article (DOI: 10.1016/j.phrs.2022.106620). This citation provides references to the earlier studies which are omitted in the current report and should be included for the reader.*

The suggested review is now cited in the appropriate part of the manuscript.

5. *In keeping with prior reports by Dorit Ron and more recently, Olsen and coworkers on noribogaine's effect on GDNF expression, administration of oxa-noribogaine (40 mg/kg; i.p.) significantly increased GDNF protein levels compared to control group. However, the magnitude compared to noribogaine was massively larger (100 – 200% increase). The authors suggest that targeted elevation of GDNF in dopamine regions of the brain underscores the improved efficacy compared to the lead molecule. Aticaprant blocks the elevation consistent with a kappa mediated effect. This part of the study is very important for defining a novel MoA for iboga alkaloids and effects of ibogaine in humans undergoing detoxification from opioids and other drugs. These studies alone warrant expedited publication of their work.*

We agree and thank the reviewer for their supportive statement.

6. *One potential concern is potential for GDNF mediated neurotoxic effects in the cerebellum. The authors do not state if the cerebellum was examined in their study (perhaps the authors have kept the brains). Comment.*

GDNF protein levels in the cerebellum were not examined as part of the current study. Unfortunately we do not have the brain samples. This issue however will be examined in the future studies (thank you).

7. GDNF has been linked to cerebellar toxicity (Luz et al., 2016) in keeping with the original work of Mark Moliver and others who demonstrated toxicity to cerebellar Purkinje neurons following high doses of ibogaine. The Purkinje cells express GDNF. This observation may warrant some discussion by the authors since as they know, the GDNF milestone started as a hypothesis for Parkinson's disease, too.

Yes, indeed. We do not observe overt signs of cerebellum-related symptoms, but this point deserves attention in the future studies. We added a note to this effect in the Limitations section.

8. Since nigrostriatal GDNF overexpression induces a robust weight loss in both animal models and clinical trials, the authors should comment on whether there was an observed food intake given the long-lasting effects of single dose administration in their animal models. Targeted expression up to two-fold increases may lead to ventricular spillover. A potential off target effect paragraph should be added in the discussion which includes GDNF.

Effect of noribogaine and oxa-noribogaine on food operant behavior is presented in Fig. 5c; there are no lasting effects on food intake. No weight loss was observed in preliminary experiments (data not shown). We added a comment in the Limitations of the Study about this concern.

9. The lack of pro-depressive or aversive behavioral effects are not induced by oxa-noribogaine is an important observation and strongly supportive of the authors' conclusions and enthusiasm overall for this class of new compounds. However, this reviewer was unclear what accounts for this improved tolerability (e.g., GDNF – dopamine vs. serotonin reuptake). This is not overly clear. Multi-target vs. single KOR-GDNF mechanism of action.

The exact mechanism for the lack of pro-depressive or aversive behavioral effects is not known at this point. We provide the partial KOR agonism hypothesis in the manuscript. Namely, the oxa-iboga compounds have sufficient signaling efficacy upstream (KOR-G protein activation) to actuate downstream PD effects like antinociception or GDNF protein elevations. At the same time, however, the KOR signaling efficacy is not high enough for the expression of the typical KOR adverse effects. While speculative, it is a reasonable leading hypothesis. In the Discussion section we also provide an alternative hypothesis, where the interaction between KOR and the background iboga pharmacology leads to the atypical behavioral profile.

Ibogaine was likewise reported to lack any CPP/CPA effects in rats (Parker, L. A., Siegel, S. & Luxton, T., 1995, Experimental and Clinical Psychopharmacology, DOI: 10.1037/1064-1297.3.4.344) or mice (Henriques, G. M. et al., 2021, Frontiers in Pharmacology, DOI: 10.3389/fphar.2021.739012). Our intention was to highlight that even significantly potentiating the KOR activity in the iboga scaffold did not result in aversive behavior, that is a traditional hallmark of potent, efficacious KOR agonists.

10. The long lasting pharmacodynamic effect of oxa-noribogaine makes this reviewer wonder if the drug demonstrates pseudoirreversible binding to KOR. Was this tested or under review?

This possibility was not investigated at this time in detail; however, the analgesic effect driven by KOR activation decreases in time, matching the PK profile.

11. This reviewer is intrigued by the pharmacokinetics of the compounds and the methods need more details on the sampling times and rationale for selection of s.c. and i.p. doses. Is there more known from dose range finding studies that are not included in this report?

Sampling times were selected to capture early exposure after administration of drug as well as full clearance time. Additional data point was included in the rat PK study due to higher dose administered as well as slower clearance in this rodent model. Routes of administration were selected in rats (I.P.) for correlation with other studies with ibogaine, noribogaine and other iboga compounds. The S.C. route in mice was used for practical reasons such as direct comparisons of iboga compounds to controls such as classic opioids and psychedelics.

No other dosages other than reported were examined in pharmacokinetic experiments.

12. What is the inferred metabolic pathway for this class? Is there an active metabolite which accounts for the rapid clearance in plasma and long aftereffects? Has this been examined in vitro?

No metabolite profiling was carried out at this point. Further, no suitable metabolic site was identified on the oxa-noribogaine scaffold yet, that would result in the formation of a stable, likely biologically active molecule. These goals will be pursued as part of the drug development studies. A note to this effect was added in the Limitations section.

13. Is oxa-ibogaine converted to oxa-noribogaine in vivo? Or does the oxygen substitution decrease CYP2D6 metabolism? Figure 3 shows rapid clearance from plasma and brain. Wash out appears to be complete in 5 hours. All of the lasting after effects are GDNF mediated? The pharmacokinetic parameters should be better described in the methods and results linked to the time course of behavioral effects. This reviewer found it difficult to decipher from a read of the methods section.

Oxa-noribogaine derivatives were selected as leading candidates for assessment in OUD models to avoid metabolic complications that would arise from the use of oxa-ibogaine analogs. In the absence of a potential metabolite scenario, we propose the GDNF rationale for the lasting after-effects. We added a note to this effect in the Discussion and Limitations section.

14. A primary observation of the current report is that oxa-iboga compounds do not show any evidence of pro-arrhythmia risks in adult primary human heart cells. Although there is much discussion of hERG, there was discussion of actual binding to or hERG block measurements? PSP screen negative at hERG?

In the revised manuscript we provide the binding affinity of oxa-noribogaine at hERG (main text and supplementary figure S3d).

15. The discussion of compensatory effects induced by the enhanced inhibition of L-type calcium channels in this species versus human (Koenig et al., reference) is outdated. In vivo QTc prolongation mitigation by calcium channel block requires very similar affinities for hERG compared to calcium (e.g., verapamil). Ibogaine does not significantly occupy calcium channels making this explanation unlikely. However, there are no data (unless it was missed by this reviewer) which informs calcium vs. hERG channel specific effects of ibogaine or oxa-ibogaine. Comment or clarify further in the results.

We have modified the relevant discussion accordingly, providing a hypothesis for the observed lack of cardiac effects in the absence of cardiac channel profiling, which is a subject of a separate study.

16. Of note, the Glue et al., (2016) paper was done in opioid dependent patients administered noribogaine 2 hours after morphine to patients who likely had residual methadone in blood (see limitations sections of the manuscript). Thus, the slope of the fit for estimating QTc vs. ng/ml noribogaine may not be conclusive in this study in contrast to what is stated by the authors.

We thank the reviewer for this information. We added a note in the manuscript to reflect this point.

17. Shi et al., (2021; ALTEX 38(4), 636-652. doi:10.14573/altex.2103311) provides an updated review of the modeling to predict cardiotoxicity. See – figure 8 comparing ibogaine to noribogaine. Therapeutic free equivalents are somewhat difficult to estimate based on a number of factors. Figure 4 shows the data for noribogaine compared to oxa-noribogaine. Was ibogaine tested in this assay? If so, the results should be included because the lead up discussion to figure 4 describes the risk associated with ibogaine. The narrative is leading the reader to conclude that ibogaine is safer than noribogaine; however metabolic ratios and in vivo electrocardiology are an emerging and still understudied area. Thus, conclusions at this juncture should be limited to the context of the authors study design, results and their relevance for a new class of modified iboga compounds.

We now state that both ibogaine and noribogaine share hERG inhibitory activity. We modified our statement to clarify that noribogaine contributes to the adverse cardiac as the primary, long lasting metabolite of ibogaine, and serves as a direct comparison compound for oxa-noribogaine.

18. The authors state that “most of the reported adverse effects have occurred > 24 hours post ibogaine ingestion (ref. 34), noribogaine with its long circulation and large exposure appears to be the culprit of cardiac risks.” Ibogaine has saturable first pass metabolism following a single dose. A review suggests that there is a range from the reported time between ibogaine ingestion and death of approximately 24 hours (range 1.5–76 hours;

Luz and Mash, 2021). High doses of ibogaine, stacked and impure drug substance along with other factors which confer risk (K⁺ and Mg⁺⁺ deficits) are unknowns.

Thank you. We removed the >24 hours statement.

19. The time course for most of these cases are uncertain and blood levels and toxicology were not available. See also ref. 14 the report from Ćimović et al. illustrates the difficulties when assessing the causality of ibogaine fatalities based on incomplete information from forensic and toxicologic investigations. Very high ibogaine concentrations were measured in femoral blood (3,260 ng/mL) and urine (28,870 ng/mL) and died 5–12 hours after oral ingestion. Thus, a definite causality statement for assigning toxicity to the metabolite or parent drug seems to this reviewer to be outside the scope of the current study. The lack of any effect in the model assay for oxa-iboga alkaloids is relevant for future drug design to limit off target effects.

We agree and limit our discussion to the fact that noribogaine as a major metabolite likely contributes to the cardiac risks of ibogaine.

Reviewer #4 (Remarks to the Author):

Ibogaine has had an interesting and important history for potential treatment of SUD but use has been hampered by cardiac toxicity. It has been recognized that Ibogaine has many psychotropic targets that include kappa opioid receptors as well as many other receptors and channels with diverse psychoactive inferences. Using innovative synthetic approaches, the authors present a large body of work characterizing the pharmacology of the novel ibogaine derivatives, oxa-noribogaine and epi-oxa. Oxa-noribogaine showed greater efficacy at the kappa opioid receptor (KOR) than noribogaine and greater efficacy than noribogaine in suppressing morphine self-administration. Importantly, unlike noribogaine (10 μ M), oxa-noribogaine (10 μ M) did not show pro-arrhythmia risks in human adult primary heart cells. Oxa-noribogaine (10 mg/kg) in rats reduced self-administration and reinstatement to both morphine and fentanyl, and unlike cocaine (10 mg/kg) and morphine (20 mg/kg), did not induce conditioned place preference at 10 and 40 mg/kg doses in mice. In addition, oxa-noribogaine, showed comparable analgesic effects to a full KOR agonist in mice, without reducing locomotor activity in the open field test. The novel compound increased glial-derived neurotrophic factor (GDNF) protein in the mPFC and VTA, but not the NAc.

Concerns:

1. Ibogaine is a promiscuous drug with regard psychoactive targets and though kappa agonists are hallucinogenic and oppose reward in many assays, care in presumption that the kappa activity is mediating the effects relevant to SUD should be taken.

We agree. Consequently, we added a comment to this effect in the Limitations section.

2. The findings in the manuscript are limited by the fact that there are discrepancies between the doses and drug comparisons across the various experiments and between

mice and rats. This prevents generalizability of the findings across the experiments. For instance, the finding that oxa-noribogaine (40 mg/kg) suppresses morphine intake in rats and 10mg/kg suppresses relapse is uncoupled from the finding that oxa-noribogaine (5.4 mg/kg) in mice does not suppress locomotion. Hypolocomotion/sedation is a major problem with interpreting behavioral assays following kappa agonist treatments and could explain a portion of the data. This discrepancy needs to be carefully addressed throughout the manuscript and perhaps an oxa-noribogaine dose response for sedation and analgesia in both mice and rats would help alleviate this potential confound.

We appreciate this point. In the revised manuscript we included a dose-response study of novelty-induced locomotion in mice, which includes 10 mg/kg (which reduces the locomotion in mice, supplementary Fig. S4b-d). However, we would like to point out that the reduction in novelty induced locomotion (or sedation) is a general non-specific readout, which does not constitute an adequate control for potential behavioral confounds in operant tests. As we note in the manuscript, still mice under the influence of oxa-noribogaine respond to external stimuli. Further, the more relevant control experiment for the oxa-noribogaine-induced suppression of opioid intake and opioid seeking reinstatement is the food operant responding in rats. While a dose of 10 mg/kg has a strong hypolocomotion effect in mice, the same dose in rats has only a small effect on food responding (in contrast to strong suppression of opioid intake). At higher doses (e.g., 40 mg/kg), the inhibitory effect on food intake is transitory, while suppression of opioid intake is long-term. Thus we can compare the effects of 10 mg/kg across behavioral paradigms in mice and rats, and most pertinently we have comparison of 10 and 40 mg/kg in rats in both opioid and food operant responding.

Nevertheless, we added a note in the Limitations section about the potential confounds due to behavior suppression in acute phases of the experiments.

3. Though there was a substantive kappa-dependent increase in GDNF protein levels in the VTA from oxa-noribogaine, what this means for which of the behavioral sequelae was unclear and not tested. The consequence of up or down regulation of GDNF in the VTA seems drug-specific and there are many questions in this reviewer's mind as to how this finding provides mechanistic substance to the paper. Some additional context re interpretation of the regional specificity in regulation of GDNF would help the reader.

We added additional context to clarify the reason why we focused on GDNF.

"We next examined down-stream molecular signaling targets, focusing on neurotrophic factors as mediators of neuroplasticity and potentially long-lasting therapeutic effects. We centered on glial cell line derived neurotrophic factor (GDNF, Fig. 2a, f) as it has been implicated in ibogaine's and noribogaine's suppression of alcohol intake in drinking rats²⁹, and shown to be induced dose-dependently by ibogaine in wild type unconditioned rats.³⁰ Although this neurotrophic factor plays complex roles in the neurobiology of substance use disorders,^{31,32} targeted elevation of GDNF protein levels in specific brain areas, such as the ventral tegmental area (VTA), enables attenuation or reversal of various aspects of addiction-related effects (e.g. drug intake, reward expression, and dopamine neuron firing patterns)."

Although the current work does not address the role of GDNF in the number of behavior readouts, the cited previous work has demonstrated, for example, that GDNF injection in the VTA reduced alcohol consumption (see the excerpt above). In our view these kinds of mechanistic studies are beyond the scope of this already extensive study. However, we believe that with the provided expanded context, the GDNF data adds to the pharmacological characterization of the new oxa-iboga compounds (downstream signaling effects) and provides a concrete hypothesis for future studies. We added a note to this effect in the Discussion and Limitation sections.

4. The oxa-noribogaine analgesia, unlike the Epi-Oxa appears to have a mu and kappa component based on the KO as well as the antagonist data in supplementary. This makes pharmacology complex. Does oxa-noribogaine have sedation at an EC80 for analgesia in the Mu KO mouse? Also, for completion it would be important to see the effects on oxa-noribogaine analgesia in the females in the Mu KO mouse (Fig 3e) since in most mouse strains females are considerably less sensitive to kappa agonist analgesia than males. Lastly, that kappa-agonist induced analgesia could have a stress-induced component should be discussed— this would not be a “desirable therapeutic-like effect” as indicated in the manuscript.

Please see the new data and proposed hypothesis for the observed pharmacological differences between oxa-noribogaine and epi-oxa-noribogaine in the point 5 below. Sedation is a non-specific readout with limited relevance for the OUD-related behavior assays as discussed above. We therefore did not pursue detailed examination of the sedation effects in KO mice (qualitative observations show no obvious differences between genotypes). Regarding the stress effects of KOR agonists, we added a note to in the main text to address this point. As suggested by the reviewer we completed the analgesia study in female KO models and observed minor differences in comparison to male KO mice. The relative order of contribution of KOR and MOR is the same (KOR as the major target), but the differential between these two contributors is smaller in females.

5. Unconvincing is that oxa-noribogaine is working as an atypical kappa agonist as opposed to a kappa/mu agonist/PAM. This possibility is based upon the overlapping kappa properties of oxa-noribogaine and epi-oxa in the invitro kappa assays yet very different in vivo pharmacological properties (sedation and analgesia). Also the analgesia data show a mu component for oxa-noribogaine but not epi-oxa in the KO data and in fig S6, which shows the mu antagonist more effective than kappa antagonist at blocking oxa-noribogaine analgesia. Finally there looks like a strong tendency for oxa-noribogaine to be rewarding in the CPP assay at 40mg/kg that statistically may not be different from the morphine results. Pure kappa agonists would be aversive in the CPP assay and I would assume epi-oxa is aversive though this data was not found in the manuscript.

To address the point about the differences in analgesia mechanism between the two compounds, we obtained new *in vitro* MOR signaling data. To recap, MOR does not contribute to the antinociceptive effects of epi-oxa in mice in contrast to oxa-noribogaine where MOR plays a minor role. Both compounds have comparable binding and signaling potency and efficacy at KOR, as well as comparable binding and signaling potency at MOR *in vitro*, but epi-oxa is markedly less efficacious in the MOR BRET assays and

inactive (below detection limit) in the MOR BRET Nb33 sensor assay (Supporting Fig. S5). Thus, we suggest that there is a signaling threshold for the upstream signaling effect to translate into an *in vivo* pharmacodynamic effect like antinociception. In the revised manuscript, we have added this new data, hypothesis, and discussion, which include limitations, and alternative hypotheses. We now provide a reasonable explanation for the different analgesia mechanism between the two compounds.

Regarding the CPP assays, the statistical tests show no significance.

6. It would be important to know if oxa-noribogaine has any affinity for the ORL-1 receptor since ORL-1 agonists can modify rewarding actions. Oxa-noribogaine was tested in the primary screen shown in a supplementary figure but hazy as to if there was significant activity/binding.

We addressed this point, no binding affinity for ORL-1 (hNOP) was detected up to 10 μ M (Supporting Fig. S3e and Table S1), a note was added in the main text.

7. The discussion dismisses sex differences in oxa-noribogaine behaviors which should be cautioned – many kappa-regulated behaviors are markedly different between sexes though not so dramatically with mu.

In response to this concern, we edited and tempered the discussion on the sex differences.

Minor:

1. Consider removing the vague, “radically different,” from the first sentence of the abstract, so that it simply reads “...prototypes of treatment of substance use disorders...”

We appreciate the sentiment. However, our long-standing research in the ibogaine field leads us to an interim conclusion that ibogaine does not simply represent “a prototype for treatment of SUDs”. It is a “dramatic” departure from the existing approaches. We tempered the “radically different” to “markedly different” as a suggested compromise.

3. The final sentence of the first paragraph on line 100, beginning with “To further probe” was unspecific and difficult to interpret until the next paragraph was read. To improve, consider more specific language for this sentence, such as “To determine whether the KOR signaling initiated by oxa-noribogaine is comparable to a full KOR agonist, we examined... which indicated...”

Done

4. The section header on line 190 omits sedative effects. This adds to some confusion once the paragraph on sedation comes up, especially as the leading sentence leads with “hallucinosi” which is misleading to the reader as to the anticipated content within the paragraph.

The sedation effects are dose dependent as discussed in the said paragraph, and thus stating “no sedation” in the section header would be an overstatement. Instead, we added aversive effects to the introductory sentence of the sedation paragraph.

5. In all figures the drug doses in mg/kg for all drug conditions should be included for each figure panel and in the figure legend – the species used should be clear.

The doses were added, each graphic focuses on one species, now indicated in the graphic legend titles and in the legend themselves.

6. The inclusion of imipramine in Fig. 3i could be further explained. The drug is mentioned just once as a monoamine reuptake inhibitor in the Introduction. Is this drug and dose intended to result in the same “pro-depressive-like effects” found after standard kappa psychedelics as described in Discussion?

Explanation was added.

REVIEWERS' COMMENTS

Reviewer #2 (Remarks to the Author):

The authors have done an excellent job addressing previous concerns. This manuscript is ready for publication.

Reviewer #3 (Remarks to the Author):

The authors have been responsive to the comments and queries of this reviewer. Additional data have been submitted to address some of the questions raised in the prior review regarding kappa and mu binding properties of oxa-noribogaine. Additional study data on MOR signaling efficacy using two different functional readouts in two species (mouse and human) strengthen the revised submission. Limitations and alternative explanations are provided in the discussion in response to many of the combined comments of the reviewers.

Based on SAR studies for this series of compounds, the authors have identified the benzofuran substitution of the indole as important for enhanced KOR binding and signaling. The KOR mechanism of action for oxa-noribogaine is a primary focus of the paper and builds on the original published kappa pharmacology of noribogaine as a partial biased agonist at this receptor.

As discussed in the rebuttal, both noribogaine and oxa-noribogaine are equivalent at the KOR pharmacophore while oxa-noribogaine has significantly increased KOR activity compared to noribogaine and higher KOR binding affinity, G protein activation efficacy via 35SGTPgammaS and BRET assays.

The authors suggest that “the observed differences in binding potency are due to electronic effects, namely changes in the electron density distribution in the ligand ring and resulting interactions with larger surfaces of the receptor.”

This discussion suggests (as stated in the prior review) the possibility of pseudo-irreversible – inactivating orthosteric site binding by oxa-noribogaine or possible allosteric site interactions (mu receptor/kappa dimers) for this new compound as a likely explanation. Thus, the mechanistic underpinnings remain uncertain and future exploratory studies will shed more light on which mechanism(s) is most relevant for understanding the robust in vivo efficacy demonstrated for the compound. This fact does not detract from the overall high enthusiasm for the novel findings reported in the current version of the manuscript.

The authors provide additional data for hERG using a radioligand displacement assay. These results are quite interesting as oxa-noribogaine and epi-oxa-noribogaine showed no pro-arrhythmic potential at any concentration tested (up to 10 μ M) as reported in the previous version of the manuscript. However, the new data added demonstrate that oxa-noribogaine binds to hERG with the same potency range as noribogaine ($K_i(\text{oxa-noriboga}) = 2.1 \mu\text{M}$, Supplementary Fig. S3d vs K_i

(noriboga) = 2.0 μ M).

The authors suggest the observed difference in pro-arrhythmia risks between the oxa-iboga and iboga compounds are related to the activity at multiple ion channels in the absence of any data to support this conclusion. They suggest that future work is necessary and beyond the scope of their report.

Indeed, future advancement of oxa-noribogaine or another lead candidate to Phase 1 will require a CIPA-Proarrhythmia (GLP) study which includes functional activity at human L-type calcium and sodium channels in addition to the hERG channel.

This reviewer finds these results intriguing as it is well known that binding affinity at hERG significantly correlates with QTc prolongation and occurrence of early after depolarizations in functional assays using cardiomyocytes. One would not expect there to be significant differences between oxa-noribogaine and noribogaine binding activity at the L-type channel to offset functional hERG effects. Hence, what could explain this important observation? Noribogaine is known to have negligible (e.g., very low) affinity at the L-type calcium channel (Koenig et al., 2013).

The authors attempt to address this important “disconnect” between hERG binding affinity (using the radioligand assay method) vs. functional hERG block by considering that “only male human hearts were used in this study (pg.12 lines 338-347)” This explanation is unclear in this context. Would females be more at risk?

Are the authors suggesting that direct KOR effects in the heart negate hERG block? Why, if hERG affinity is similar for both molecules, would there be a significant lack of QTc prolongation for oxa-noribogaine but not noribogaine? The purported safety of this novel class of molecules is a major conclusion of this study. This reviewer suggests that the additional data on hERG in the absence of a broad cardiac panel using well validated functional assay may require a more nuanced discussion of safety and a bit more discussion on possible mechanism.

One possibility is capacity, consistent with activation-state dependent binding by the benzofuran series which may be different compared to noribogaine. Temperature and ionic effects are important in cardiomyocyte functional assays which could change binding site affinities. Since they have not conducted a GLP study, there is no clear explanation. However, if the authors believe direct kappa receptor mediated effects are relevant then they should provide further explanation with cited references.

Kappa-active compounds have cardiovascular effects albeit their results are mixed. In rat studies, kappa opioid receptor agonists (KORAg) are cardioprotective (Jaiswal et al, 2010) and reduce arrhythmias (Xiu et al, 2013). U-50,488H has effects on transient outward K⁺ and ultrarapid delayed rectifier K⁺ currents in human pediatric atrial myocytes that are not antagonized by nor-BNI or naloxone (Xiao et al, 2003).

This reviewer suggests that some comment could be added to the limitation section of the

discussion regarding cardiac safety given the overlapping potency at hERG and the known lack of calcium channel activity of noribogaine that would be needed to offset decreased repolarization reserve in heart. The oxa-noribogaine compound would need to show similar affinity to L-type cardiac channels (e.g., like verapamil).

Something else must explain the authors findings from adult primary cardiomyocyte functional assays. This suggested edit and additional citations in the discussion are considered important for the reader.

Reviewer #4 (Remarks to the Author):

The authors have well-addressed the prior concerns.

Comments by reviewer #2.

The authors have been responsive to the comments and queries of this reviewer. Additional data have been submitted to address some of the questions raised in the prior review regarding kappa and mu binding properties of oxa-noribogaine. Additional study data on MOR signaling efficacy using two different functional readouts in two species (mouse and human) strengthen the revised submission. Limitations and alternative explanations are provided in the discussion in response to many of the combined comments of the reviewers. Based on SAR studies for this series of compounds, the authors have identified the benzofuran substitution of the indole as important for enhanced KOR binding and signaling. The KOR mechanism of action for oxa-noribogaine is a primary focus of the paper and builds on the original published kappa pharmacology of noribogaine as a partial biased agonist at this receptor. As discussed in the rebuttal, both noribogaine and oxa-noribogaine are equivalent at the KOR pharmacophore while oxa-noribogaine has significantly increased KOR activity compared to noribogaine and higher KOR binding affinity, G protein activation efficacy via 35SGTPgammaS and BRET assays. The authors suggest that “the observed differences in binding potency are due to electronic effects, namely changes in the electron density distribution in the ligand ring and resulting interactions with larger surfaces of the receptor.” This discussion suggests (as stated in the prior review) the possibility of pseudo-irreversible – inactivating orthosteric site binding by oxa-noribogaine or possible allosteric site interactions (mu receptor/kappa dimers) for this new compound as a likely explanation. Thus, the mechanistic underpinnings remain uncertain and future exploratory studies will shed more light on which mechanism(s) is most relevant for understanding the robust in vivo efficacy demonstrated for the compound. This fact does not detract from the overall high enthusiasm for the novel findings reported in the current version of the manuscript.

Response: We thank the reviewer for sharing their insights.

The authors provide additional data for hERG using a radioligand displacement assay. These results are quite interesting as oxa-noribogaine and epi-oxa-noribogaine showed no pro-arrhythmic potential at any concentration tested (up to 10 μM) as reported in the previous version of the manuscript. However, the new data added demonstrate that oxa-noribogaine binds to hERG with the same potency range as noribogaine ($K_i(\text{oxa-noriboga}) = 2.1 \mu\text{M}$, Supplementary Fig. S3d vs $K_i(\text{noriboga}) = 2.0 \mu\text{M}$). The authors suggest the observed difference in pro-arrhythmia risks between the oxa-iboga and iboga compounds are related to the activity at multiple ion channels in the absence of any data to support this conclusion. They suggest that future work is necessary and beyond the scope of their report. Indeed, future advancement of oxanoribogaine or another lead candidate to Phase 1 will require a CIPA-Proarrhythmia (GLP) study which includes functional activity at human L-type calcium and sodium channels in addition to the hERG channel. This reviewer finds these results intriguing as it is well known that binding affinity at hERG significantly correlates with QTc prolongation and occurrence of early after depolarizations in functional assays using cardiomyocytes. One would not expect there to be significant differences between oxa-noribogaine and noribogaine binding activity at the L-type channel to offset functional hERG effects. Hence, what could explain this important observation? Noribogaine is known to have negligible (e.g., very low) affinity at the L-type calcium channel (Koenig et al., 2013).

Response: We agree with this reviewer that it is unlikely that oxa-noribogaine has a meaningful inhibitory activity at the L-type calcium channels, by analogy to noribogaine. However, in our experience we have observed that even small structural changes in the iboga system can have unexpected and far-reaching pharmacological effects. We therefore agree that the CIPA-Proarrhythmia (GLP) panel and related studies are required to address this mechanistic question, which will be carried out in future studies. In the meantime, we would like to note that interaction with other ion channels, not just L-type calcium channel (CaV1.2), are known to compensate for hERG inhibition in terms of proarrhythmia risks, such as NaV1.5. FDA recommendation is that at minimum hERG, CaV1.2, and NaV1.5 channel studies using patch clamp method are needed to support an integrated risk assessment of drugs (www.fda.gov/media/151418/download). We added a note at the end of the cardiomyocyte section to highlight this possibility and hypothesis.

“We hypothesize that the observed differences in pro-arrhythmia risks between the oxa-iboga and iboga compounds are related to the activity at multiple cardiac ion channels of the former compounds, likely engaging channels known to compensate hERG-mediated pro-arrhythmia risks (such as the channel Nav1.5). However, to elucidate the relevant mechanism will require detailed follow-up studies, which are beyond the scope of this report.”

The authors attempt to address this important “disconnect” between hERG binding affinity (using the radioligand assay method) vs. functional hERG block by considering that “only male human hearts were used in this study (pg.12 lines 338-347)” This explanation is unclear in this context. Would females be more at risk? Are the authors suggesting that direct KOR effects in the heart negate hERG block? Why, if hERG affinity is similar for both molecules, would there be a significant lack of QTc prolongation for oxanoribogaine but not noribogaine? The purported safety of this novel class of molecules is a major conclusion of this study. This reviewer suggests that the additional data on hERG in the absence of a broad cardiac panel using well validated functional assay may require a more nuanced discussion of safety and a bit more discussion on possible mechanism. One possibility is capacity, consistent with activation-state dependent binding by the benzofuran series which may be different compared to noribogaine. Temperature and ionic effects are important in cardiomyocyte functional assays which could change binding site affinities. Since they have not conducted a GLP study, there is no clear explanation. However, if the authors believe direct kappa receptor mediated effects are relevant then they should provide further explanation with cited references. Kappa-active compounds have cardiovascular effects albeit their results are mixed. In rat studies, kappa opioid receptor agonists (KORAg) are cardioprotective (Jaiswal et al, 2010) and reduce arrhythmias (Xiu et al, 2013). U-50,488H has effects on transient outward K⁺ and ultrarapid delayed rectifier K⁺ currents in human pediatric atrial myocytes that are not antagonized by nor-BNI or naloxone (Xiao et al, 2003). This reviewer suggests that some comment could be added to the limitation section of the discussion regarding cardiac safety given the overlapping potency at hERG and the known lack of calcium channel activity of noribogaine that would be needed to offset decreased repolarization reserve in heart. The oxanoribogaine compound would need to show similar affinity to L-type cardiac channels (e.g., like verapamil). Something else must explain the authors findings from adult primary cardiomyocyte functional assays. This suggested edit and additional citations in the discussion are considered important for the reader.

Response: There was a minor misunderstanding as we neither state explicitly nor imply that the observed differences between the oxa-iboga and iboga compounds could be due to sex differences

in KOR activity. Although the presence of opioid receptors in human myocardial cells was indicated via immunoreactivity of mu-, kappa- and delta-opioid receptors (Sobanski P., et al, DOI: 10.1007/s00380-013-0456-5), this finding has not been confirmed by other published studies. While agonists of opioid receptors have been reported to directly modulate cardiac electrophysiology, the lack of attenuation of these effects by the antagonists, like naltrexone and naloxone, imply this modulation is independent of the opioid receptors (Katchman LN., et al DOI: 10.1124/jpet.102.038240; Tran PN., et al DOI: 10.1371/journal.pone.0241362; Wu C., et al DOI: 10.1113/expphysiol.1997.sp004021; G. Boachie-Ansah G., et al, DOI: 10.1111/j.1476-5381.1989.tb12019.x). Instead, as discussed above, we propose that other cardiac ion channels, such as NaV1.5, may compensate for the hERG effects, as an explanatory hypothesis and guide for future studies addressing the cardiotoxicity differences between the oxa-iboga and iboga alkaloids. A statement to this effect was added in the cardiomyocyte section of the article (for more details please see the point and response above).